# Maternal gut and breast milk microbiota affect infant gut antibiotic resistome and mobile genetic elements

Katariina Pärnänen[1], Antti Karkman[2,3,4], Jenni Hultman [1], Christina Lyra[1], Johan Bengtsson-Palme [2,3,5], D.G.Joakim Larsson [2,3], Samuli Rautava[6], Erika Isolauri[6], Seppo Salminen[7], Himanshu Kumar[7], Reetta Satokari[8] & Marko Virta [1]

The infant gut microbiota has a high abundance of antibiotic resistance genes (ARGs) compared to adults, even in the absence of antibiotic exposure. Here we study potential sources of infant gut ARGs by performing metagenomic sequencing of breast milk, as well as infant and maternal gut microbiomes. We find that fecal ARG and mobile genetic element (MGE) profiles of infants are more similar to those of their own mothers than to those of unrelated mothers. MGEs in mothers' breast milk are also shared with their own infants. Termination of breastfeeding and intrapartum antibiotic prophylaxis of mothers, which have the potential to affect microbial community composition, are associated with higher abundances of specific ARGs, the composition of which is largely shaped by bacterial phylogeny in the infant gut. Our results suggest that infants inherit the legacy of past antibiotic consumption of their mothers via transmission of genes, but microbiota composition still strongly impacts the overall resistance load.

[1] Department of Microbiology, University of Helsinki, Viikinkaari 9, Helsinki 00014, Finland. [2] Department of Infectious Diseases, Institute of Biomedicine, The Sahlgrenska Academy, University of Gothenburg, Guldhedsgatan 10, SE-413 46, Gothenburg, Sweden. [3] Center for Antibiotic Resistance research (CARe) at University of Gothenburg, P.O. Box 440SE-40530 Gothenburg, Sweden. [4] Faculty of Biological and Environmental Sciences, University of Helsinki, 00014 Helsinki, Finland. [5] Wisconsin Institute for Discovery, University of Wisconsin-Madison, 330N. Orchard Street, Madison, WI 53715, USA. [6] University of Turku and Turku University Hospital, 20500 Turku, Finland. [7] Functional Foods Forum, Faculty of medicine, University of Turku, Turku 20520, Finland. [8] Immunobiology Research Program, Faculty of Medicine, University of Helsinki, Helsinki 00014, Finland. Correspondence and requests for materials should be addressed to K.Pärän. (email: katariina.parnanen@helsinki.fi)

Antibiotic resistance is one of the main global threats to public health[1]. Resistance to antibiotics is often conferred on bacteria by antibiotic resistance genes (ARGs), all of which together form the antibiotic resistome[2]. Being colonized by opportunistic pathogens carrying ARGs increases the risk of acquiring infections that are difficult to treat[3]. The risk associated with carrying resistant pathogens are especially large in infancy when the gut microbiota and immune system are less stable and developed compared to in adults[4–6]. Globally, an estimated 214,000 annual neonatal deaths are due to septic infections caused by antibiotic resistant pathogens[5] illustrating the drastic effects of the antibiotic resistance crisis on infant health.

Recent research has revealed that the infant gut has a high abundance of ARGs compared to adults even when infants have not been exposed to antibiotics[7–9]. The high amounts of ARGs carried by infants' bacteria in the absence of antibiotic selection pressure raise questions about their origin, since the selection must have occurred prior entering the infant's gut, for example during antibiotic treatment of the infant's mother. The high abundance of Gammaproteobacteria in the infant gut has been hypothesized to explain the higher levels of ARGs as Gammaproteobacteria often carry several resistance genes[10] and encompass taxa which are among the first colonizers of the gut[7–9,11,12]. However, the origin of the resistant bacterial strains to the infant gut is not known, as they can be transmitted from the environment, other individuals or the mother. Although similar ARG profiles have been found in the guts of mothers and infants, suggesting the possibility sharing of resistance genes[12,13], previous studies have reported that infants and their mothers do not have significantly more similar gut resistomes than unrelated individuals[8,12]. Breast milk—the primary source of nutrition during the first months—shapes the infant gut microbiota[14,15], but its contribution to the resistome is virtually unexplored with the exception of a study on preterm infants[16]. Moreover, mobile genetic elements (MGEs), forming the mobilome, can transfer ARGs between members of the gut microbial community[17–19], but their role in shaping the resistome in the infant gut is unclear.

In this study, we assessed whether mothers' gut and breast milk microbiota influence the infant gut resistome and MGEs during the first 6 months of life by quantifying sharing of genes and bacteria between mothers and infants. This was done using deep metagenomic sequencing of 16 mother-infant pairs over a period of 8 months, totaling in 96 breast milk or fecal samples of which half overlapped by sampling time (Supplementary Data 1–3). Our metagenomic characterization of the resistome and MGEs of breast milk is the most deeply sequenced breast milk metagenome to date[20–22]. Our results show that some of the mother's ARGs and MGEs are transferred to the infant gut from the mother's gut microbiota as well as breast milk. We also found that maternal intrapartum antibiotic prophylaxis (IAP), breastfeeding duration and bacterial phylogeny affect the resistome and MGE composition. Our study also demonstrates that breastfeeding reduces the relative abundance of certain ARGs and MGEs in infants who are breastfed for 6 months or longer compared to infants who are breastfed for a shorter time. Correspondingly, IAP of mothers increases the abundance of certain ARGs and MGEs in infants. The results suggest that mothers contribute to the infant gut microbiota's resistome and mobilome development by sharing genes from their gut and breast milk bacteria.

## Results

**Infant gut and breast milk have high levels of ARGs and MGEs.** Infant gut microbiomes had higher abundances of ARGs and MGEs than the gut microbiomes of their mothers ($p < 0.05$, negative binomial generalized linear models (GLMs) and Tukey's

post hoc test) (Fig. 1c, d, Supplementary Tables 1&2). This was apparent even though the infants had not been exposed to antibiotics during their life. However, the diversity of the MGEs and ARGs did not significantly differ between the two groups despite that the taxonomic diversity was significantly lower in infants ($p < 0.05$, analysis of variance (ANOVA) and Tukey's post hoc test, Fig. 1a, b and Supplementary Fig. 1, Supplementary Tables 3–6). The observation that infants have higher ARG abundances compared to adults has been made previously[8], but similar studies on the MGE abundance in the infant gut have not been conducted.

The relative abundances of ARGs were similar in breast milk and fecal samples from 6-month-old infants and mothers ($p > 0.05$, negative binomial GLMs and Tukey's post hoc test, Fig. 1c and d, Supplementary Table 1). Breast milk had the lowest microbial diversity of all sample types ($p < 0.05$, (ANOVA) and Tukey's post hoc test, Supplementary Fig. 1, Supplementary Tables 3&4). MGEs were significantly more abundant in breast milk than in mothers' feces ($p < 0.05$, negative binomial GLMs and Tukey's post hoc test, Fig. 1b, Supplementary Table 2). However, due to the lower sequencing depth in breast milk compared to feces, there is uncertainty in the exact estimates of both the relative abundance and diversity, which should be noted.

Principal coordinates analysis (PCoA) was done to cluster samples using relative abundance (Fig. 2, Supplementary Tables 6&7) and presence–absence data (Supplementary Fig. 2, Supplementary Tables 6&7) on the microbiomes, resistomes and MGEs. Breast milk harbored a microbial community, resistome and MGE profile distinct from the gut (ADONIS, Benjamini & Hochberg method adjusted $p$-value < 0.05, Fig. 2a–d and Supplementary Fig. 2, Supplementary Tables 7&8). Breast milk was overall more similar to the infant than maternal gut. We observed that infants and mothers had differences not only in their microbial community and resistome composition which have been observed previously[7,8], but also in their MGE composition (ADONIS, Benjamini & Hochberg method adjusted $p$-value < 0.05, Fig. 2a–d and Supplementary Fig. 2, Supplementary Table 7). Characterization of the MGEs was enabled by using a custom MGE database (Materials and methods).

Differences in the resistome compositions between adults and infants and gut and breast milk were also seen in ARG and MGE classes (Fig. 3a, b). Tetracycline resistance genes were the most abundant resistance class in mothers' feces and also prevalent in infants, which is typical for gut microbiota[10,23,24]. Interestingly, all resistance gene classes except tetracycline, MLSB (macrolide-lincosamide-streptogramin B) and trimethoprim resistance were more abundant in infants compared to mothers (negative binomial GLM, Tukey's post hoc test, adjusted $p$-value < 0.05). The colistin resistance gene class including the *pmr* and mobile *mcr* gene families was among the 12 most abundant classes and abundant especially in infants. Nonetheless, we did not detect any mobile colistin resistance genes. Transposases constituted the most common MGE class in all samples (Fig. 3). The higher abundance of ARGs and MGEs including conjugative plasmids and transposons in the infant gut (Figs. 1c, d and 3a, b) indicates that there is a higher potential for antibiotic resistance and horizontal gene transfer than in the adult gut.

Gammaproteobacteria and Bacilli were more common in infants than in mothers while mothers had more Clostridia, Bacteroides and Verrumicrobiae (Fig. 3d). On class level, Actinobacteria was most abundant in fecal samples and Bacilli in milk samples. The most abundant genus in feces of infants and mothers was *Bifidobacterium* and genus *Streptococcus* in breast milk (Fig. 3c). Genera which were significantly different (DESeq2, negative binomial GLMs, Wald's test, $p < 0.05$) between infants and mothers and 1- and 6-month-old infants using DESeq2[25]

analysis are depicted in Supplementary Fig. 3 and Supplementary Fig. 4.

**Infant gut resistomes are linked to microbiome composition.** The microbial community composition structured the resistome and MGE composition in the fecal microbiome of infants and mothers and in breast milk. The resistome, MGE and microbial community distance matrices correlated significantly with each other (Mantel's test, cor ≥ 0.5 and $p ≤ 0.001$ for all pairwise correlations). The correlation between microbial community structure and ARGs suggests that the phylogenetic composition at least partially governs the resistome and mobilome composition in the gut microbiome of infants and mothers and in breast milk, similar to what has been seen in soil habitats[26]. Interestingly, there also seems to be an association between the microbial community and MGEs. However, MGEs did not show correlation patterns with any specific taxa on species, genus, and class levels (Supplementary Data 4).

On the other hand, strong correlations between taxa, such as *Escherichia coli* and Gammaproteobacteria, and the total sum of ARG abundances were observed in the infants at both 1 and 6 months of age (negative binomial GLMs, Benjamini & Hochberg method adjusted $p$-value < 0.05, Supplementary Data 4). *E. coli* best predicted the ARG abundances in infants (model validation with $\chi^2$-test $p < 0.05$, negative binomial GLM, estimate = 0.062, $p = 6.87e-07$). This confirms the existing

hypothesis that Gammaproteobacteria contribute to high resistance load in infants[7,11] and most likely harbor the majority of the most abundant ARGs. Infants had more Gammaproteobacteria, including *E. coli*, than the mothers (Fig. 3 and Supplementary Fig. 1). Gammaproteobacteria and Enterobacteriaceae blooms have been linked to increased rates of horizontal gene transfer between opportunistic pathogens and commensals during states of gut microbiota dysbiosis[27], suggesting that there might be elevated rates of gene transfer in the infant gut with high abundances of Enterobacteriaceae. This could further promote the spread of ARGs in the infant gut. However, breastfeeding might be beneficial in reducing the ARG load, as breast milk has been shown to decrease the abundance of Enterobacteriaceae and *E. coli* in the infant gut[28,29].

Despite the fact that breast milk had similar total relative abundances of ARGs to fecal samples, the abundant taxa in breast milk samples were negatively correlated with the total relative ARG abundances in infants (negative binomial GLMs, Benjamini & Hochberg method adjusted $p$-value < 0.05, Supplementary Data 4). We observed that of all the genera, *Bifidobacterium* had the strongest negative correlation with ARG abundance in infants (Supplementary Data 4). *Bifidobacterium* was too rare to detect in the breast milk samples using metagenomics. Previous studies using 16S rRNA gene amplicon sequencing, which enables more sensitive characterization of taxonomic information, have reported that the abundance of *Bifidobacterium* in breast milk

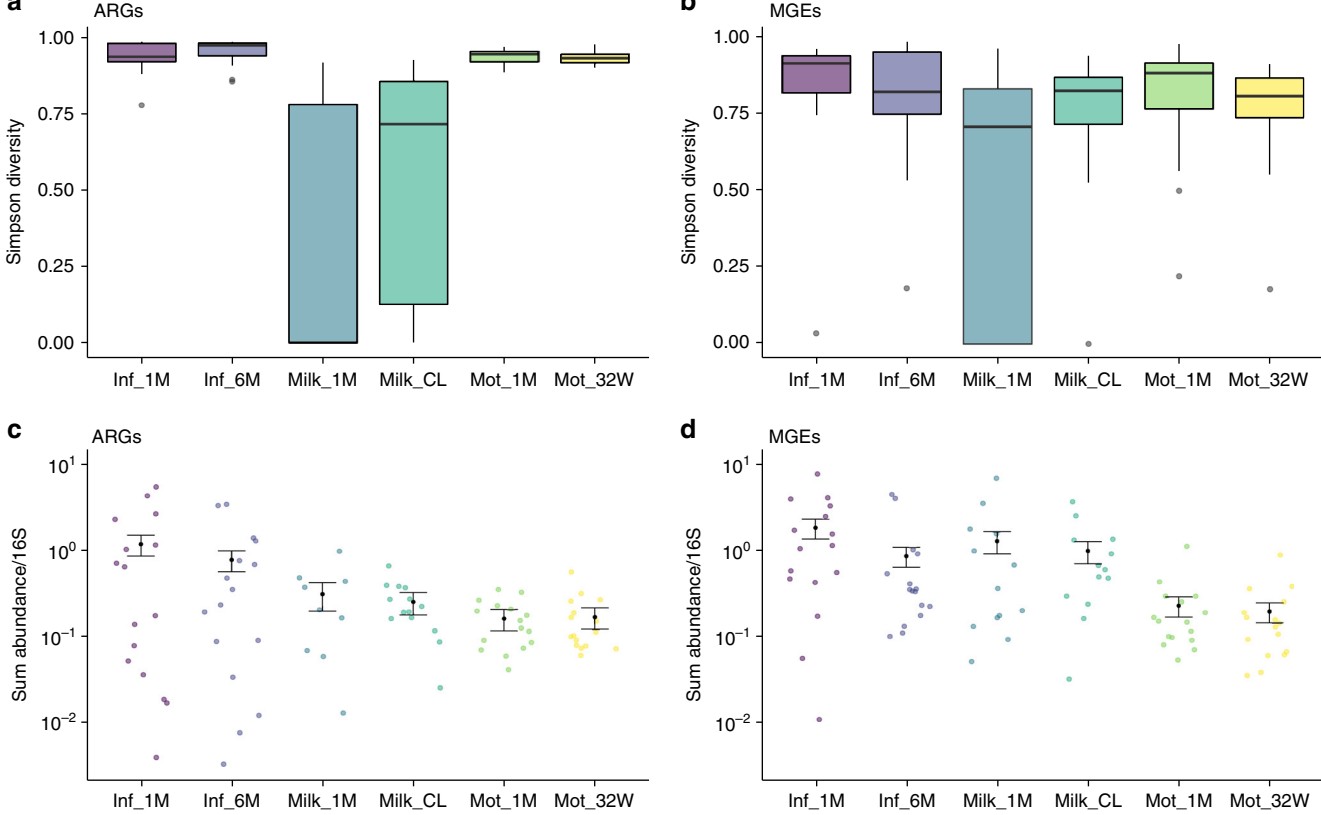

**Fig. 1** Simpson diversity of ARGs and MGEs and their relative sum abundances. **a** ARG diversity. **b** MGE diversity. **c** ARG sum relative abundances. **d** MGE sum relative abundances. The relative sum abundances are calculated per copy of 16S rRNA gene and normalized by gene lengths. Each colored point represents one sample. Predicted mean and standard error from the negative binomial GLM is drawn in black. Sample names are as follows: Inf_1M = 1-month-old infants, Inf_6M = 6-month-old infants, Mot_32W = mother fecal samples gestational week 32, Mot_1M = mother fecal samples one month postpartum, Milk_CL = colostrum or milk produced within 7 days after delivery, Milk_1M = milk 1 month postpartum. Each sample type has an *n* of 16. In boxplots **a** and **b**, the lower hinge represents 25% quantile, upper hinge 75% quantile and center line the median. Notches are calculated with the formula median ± 1.58 * interquartile range/sqrt(*n*). Negative binomial general linearized models (GLMs) were used to predict the means and standard errors (SEs) of the relative sum abundances in different sample types. The notches in **c** and **d** represent SEs and means are represented as black points

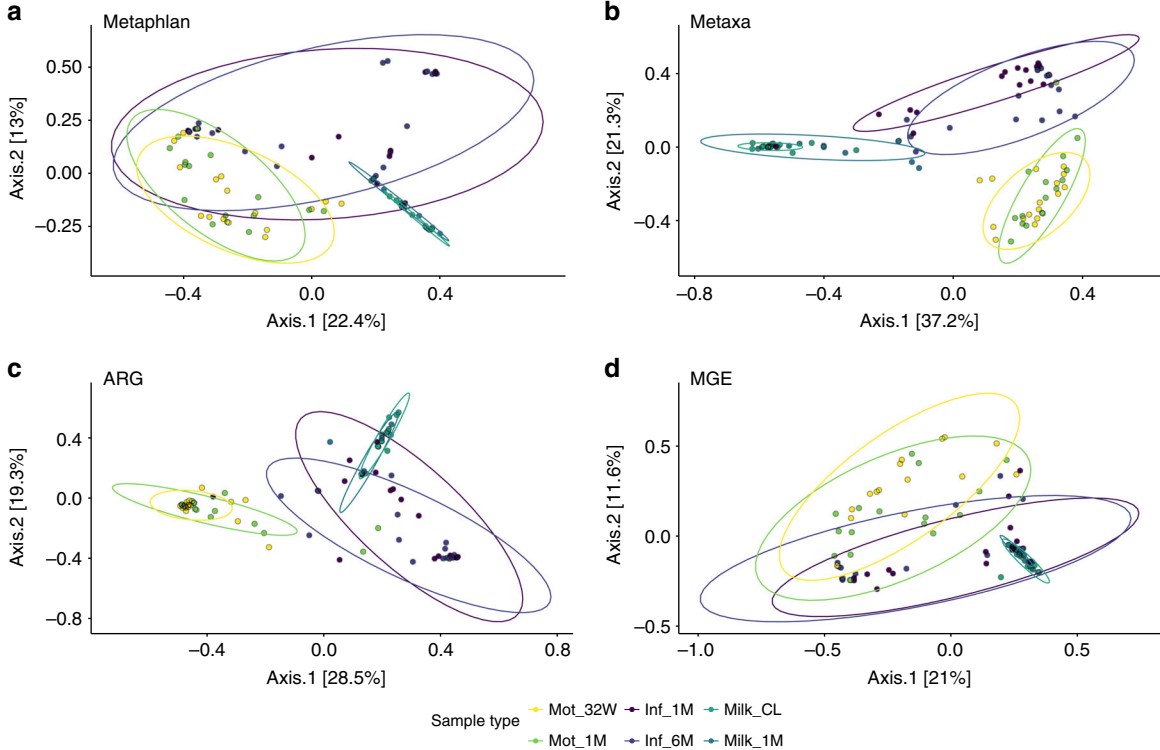

**Fig. 2** PCoA of microbiomes, resistomes, and MGEs using relative abundance. **a** Species level taxonomic identification done based on single copy marker genes with Metaphlan2[44]. **b** Taxonomic profiling based on 16S rRNA reads retrieved using Metaxa2[41]. **c** Resistome profiles based on reads mapped against an ARG database and normalized to 16S rRNA gene reads and gene lengths. **d** MGE profiles based on read mapping against a custom MGE database. Horn-Morisita similarity indexes were used to calculate between sample overlap for the ordinations. The confidence ellipses are drawn with confidence level of 0.90. Sample names are as follows: Inf_1M = 1-month-old infants, Inf_6M = 6-month-old infants, Mot_32W = mother fecal samples gestational week 32, Mot_1M = mother fecal samples 1 month postpartum, Milk_CL = colostrum or milk produced within 7 days after delivery, Milk_1M = milk 1 month postpartum. The significances and $R^2$-values of differences between samples are represented in Supplementary Tables 6 and 7

ranges from undetectable to 1.3%[15,30,31]. Even though *Bidifobacterium* is a minor member of the breast milk microbial community, the abundance of this genus increases in the infant gut as a consequence of breastfeeding and human milk oligosaccharides[14] and it was the most dominant genus in the infant gut (Fig. 3c). Our results indicate that colonization by *Bifidobacterium* might be beneficial for reducing the resistance gene load in the infant gut.

**Infants' ARGs and MGEs resemble those of their mothers.** Infants shared 40% of their ARG and 37% of their MGE types in their guts with their mothers and correspondingly, 20 and 12% with breast milk (Supplementary Fig. 5). Of the ARGs detected in breast milk, 70 and 46% overlapped between infant and maternal gut, respectively. It is likely that some of the genes shared between breast milk and infant gut are directly transferred via breast-feeding to the gut by shared species, as 76% of the species found in breast milk were also detected in the infant gut. This indicates that while several genes are common between mothers and infants, some ARGs and MGEs might be acquired by the infant from other maternal body sites such as the skin and mouth or from the urogenital tract. Some of the bacteria are likely also acquired from non-maternal sources as has been suggested previously[8].

The infant gut microbial community, resistome and MGE compositions were significantly more similar to each infant's own mother's gut microbiota than to unrelated women (ANOVA, family, $p < 0.05$, Fig. 4a–f for microbiota, resistomes, and MGEs, respectively). The results indicate similarity of microbial community composition, ARGs and MGEs between related mothers and infants. Previous studies have not been able to pinpoint that infants share significantly more ARGs with their mothers than with unrelated adults possibly due to small set of subjects or methodological challenges[8,12]. Our results suggest that maternal ARGs and MGEs have significant effects on the resistome and MGE composition of the infant gut and that there likely is transfer of some of these genes between mothers and infants.

When we compared the infant gut to breast milk, we observed that the infant gut MGE type and DNA sequence profile compositions were more similar to those of each infant's own mother's breast milk than to milk from unrelated mothers (ANOVA, family, $p < 0.05$, Fig. 5e, f), indicating that MGEs are shared between mothers and infants also via breast milk. Correspondingly, the mothers' breast milk MGE DNA sequence profiles were more similar to DNA profiles of the mother's own feces than to the fecal DNA profiles of other mothers (ANOVA, individual, $p < 0.05$, Supplementary Fig. 6). The similarities of MGEs in the gut and breast milk of related mothers and infants (ANOVA, family, $p < 0.05$, Fig. 5e, f) and breast milk and feces from the same mother (ANOVA, individual, $p < 0.05$, Supplementary Fig. 6) were observed despite the vastly different conditions in gut and breast milk. Interestingly, the sharing of MGE types and nucleotide profiles between gut and breast milk was not reflected in the taxonomic compositions and resistomes (ANOVA, family, $p > 0.05$, Fig. 5a–d and individual, ANOVA, $p > 0.05$ Supplementary Fig. 6d, e). However, this might be due to limitations in the sequencing of breast milk microbial DNA resulting in a lower sequencing depth, making it more difficult to observe sharing of low-abundant species and genes.

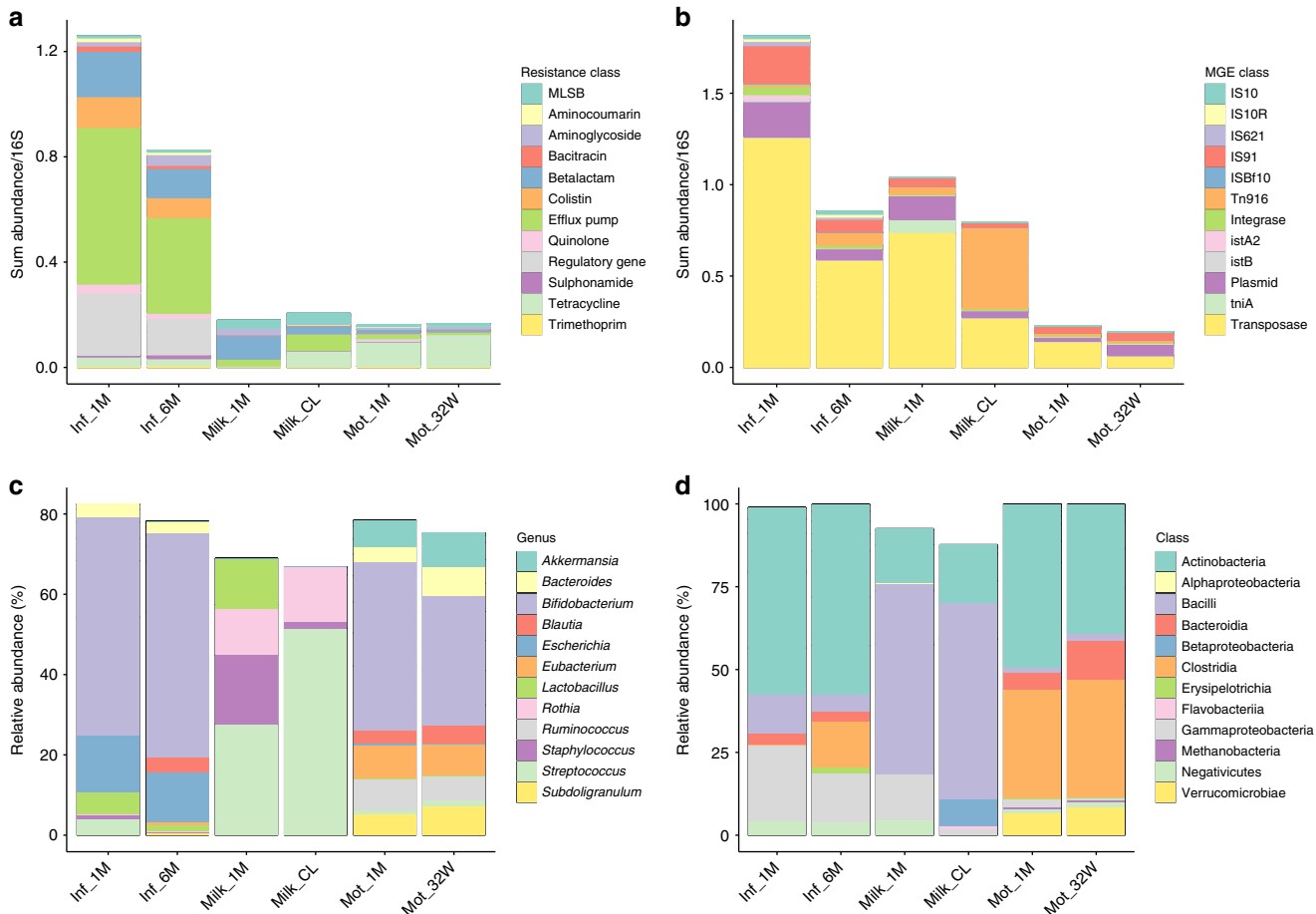

**Fig. 3** Abundant bacteria, ARGs, and MGEs in breast milk and infants' and mothers' gut. **a** Most abundant resistance classes **b** Most abundant MGE classes. **c** Most abundant genera based on Metaphlan2[44] taxonomy profiling. **d** Most abundant classes based on Metaphlan2[44] taxonomy profiling. ARG and MGE sum abundances are normalized to 16S rRNA gene as in Fig. 1 and depicted on the y-axis in **a** and **b**. The mean relative abundances for taxa, expressed as percentages, are depicted on the y-axis in **c** and **d**. Sample names are as follows: Inf_1M = 1-month-old infants, Inf_6M = 6-month-old infants, Mot_32W = mother fecal samples gestational week 32, Mot_1M = mother fecal samples 1 month postpartum, Milk_CL = colostrum or milk produced within 7 days after delivery, Milk_1M = milk 1 month postpartum

Overall, the infant microbial communities, resistomes and MGEs on species and gene type level were more similar to other infants than to those of their mothers (ANOVA, same type vs. same family, $p < 0.05$, Fig. 4a, c, e). This is likely due to differences in the gut environmental conditions between infants and their mothers being reflected in the microbiome and, by consequence, also the resistome, as it seems that phylogeny is a major determinant of the gut resistome. However, on DNA sequence profile level, the infant total microbial communities were significantly more similar to their own mothers than to other infants (ANOVA, same family vs. same type, $p < 0.05$, Fig. 4b). This result suggests that the signature of microbial strains transmitted from mothers is more pronounced than the signature related to infancy or adulthood, despite there being differences in the microbial community composition between infants and mothers (ANOVA, type, $p < 0.05$, Fig. 4a, b).

We observed that there were no significant differences in the similarity of infants and their mothers in their microbial community, resistome or MGEs in relation to sampling time overlap versus different sampling times (ANOVA, $p > 0.05$, Supplementary Fig. 7) suggesting that the familial signature is stable for several months. This might be the result of a constant transmission between family members or due to stabilization of strains and genes in the infant gut. Correspondingly, the infant

microbial community, resistome and MGE compositions were significantly more similar to their own than to an infant of the same age (ANOVA, $p < 0.05$, Supplementary Fig. 6) suggesting that the signature of each infant is more pronounced than the signature related to the infant's age. The results indicate that the transmission signature of each family or individual, which was observed in the microbiomes, resistomes, and MGEs, is persistent.

Since half of the mothers received IAP antibiotics, we investigated whether the antibiotics had an effect on how similar the mothers' and infants' microbiomes, resistomes, and MGEs were. We did not observe any differences in similarities between related infants and mothers in microbial communities, ARGs, and MGEs based on antibiotic treatment (Supplementary Fig. 8). However, IAP group's mothers and infants shared significantly less species than the control group (ANOVA, $p < 0.05$, Supplementary Fig. 8).

**IAP and terminating breastfeeding increase ARGs and MGEs.** We studied the possible contribution of breastfeeding duration and intrapartum antibiotic prophylaxis (IAP) on the development of the infant gut microbiome, resistome, and MGEs. Termination of all breastfeeding before 6 months of age, which is the recommended duration of exclusive breastfeeding according to the

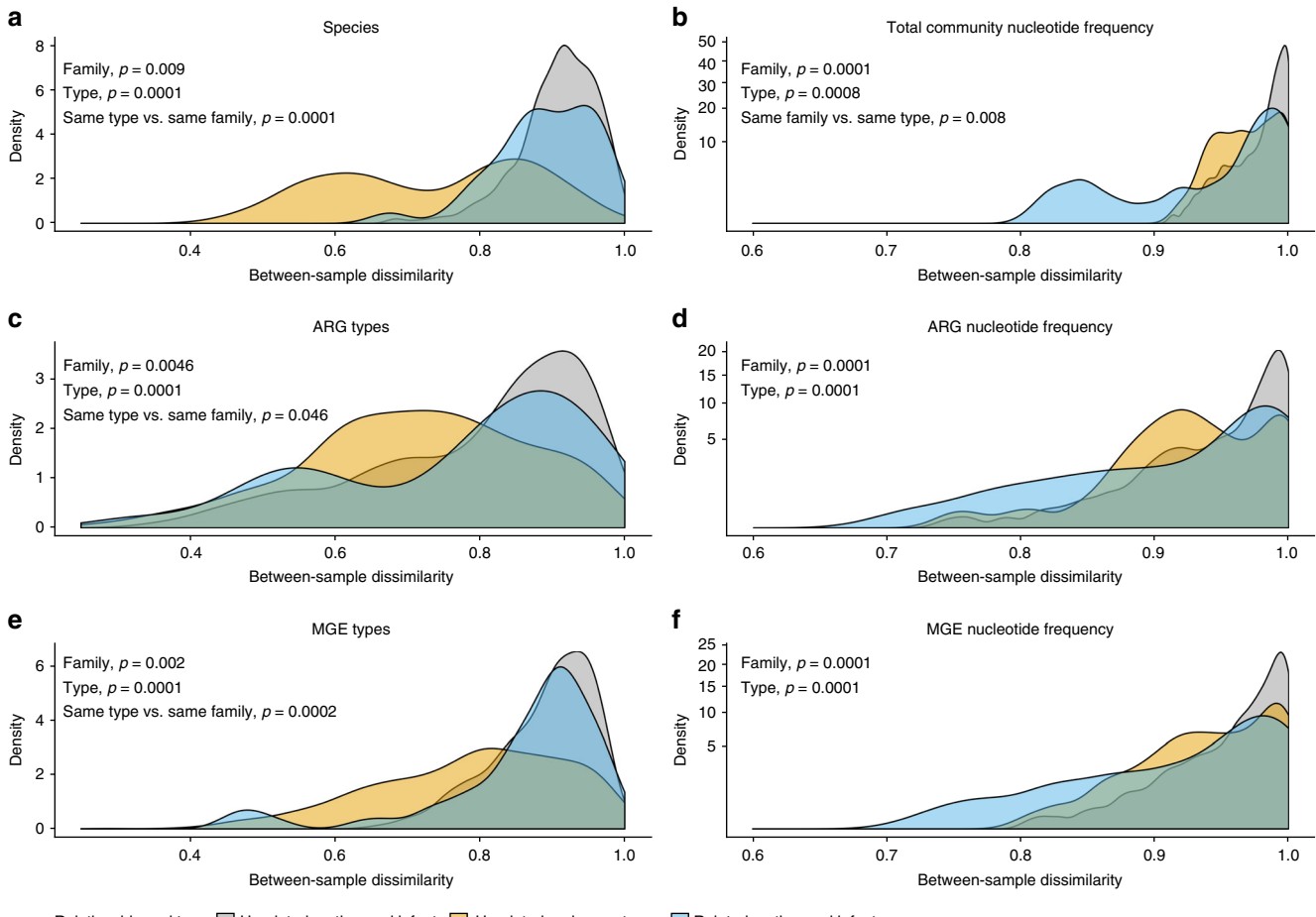

**Fig. 4** Dissimilarities of infant and mothers' gut microbiota, resistomes, and MGEs. **a** Dissimilarity of microbial community on species level between infants and mothers using Metaphlan2[44] species classifications. **b** Dissimilarity of microbial communities in infants and mothers using DNA sequence profiles calculated based on kmer profiles. **c** Dissimilarity of resistomes between infants and mothers using gene type annotations. **d** Dissimilarity between resistomes using DNA sequence profiles of genes calculated based on kmer profiles. **e** Dissimilarity of MGEs between infants and mothers using gene type annotations. **f** Dissimilarity between MGEs using DNA sequence profiles of genes calculated based on kmer profiles. Dissimilarities between related and unrelated infant-mother pairs were compared. Type notion indicates that mothers and infants are significantly different from each other, family indicates that infant's feces are more similar to mother's feces than to that of unrelated women, same type vs. same family indicates that infants are more similar to each other than to their own mothers. Significance of differences was tested using ANOVA between the similarity indexes in the comparisons and p-values < 0.05 are indicated in the figures. The density plot depicts where comparisons between sample pairs are located on the dissimilarity scale. The higher the density is at a given dissimilarity, the more pairwise comparisons have the given dissimilarity value. Density of the samples is plotted on the y-axis and the x-axis depicts the between sample Jaccard similarity index of species, ARGs and MGEs shared between sample types using presence–absence data or DNA sequence profiles based on kmers calculated with sourmash[56]. DNA sequence profiles provide a way to compare DNA sequence signatures of samples with each other and does not rely on species or gene annotations. The x-axis ranges from 0 (no dissimilarity, i.e., completely similar) to 1 (complete dissimilarity)

World Health Organization[32], was associated with an enrichment of several ARGs and MGEs in infants (negative binomial GLMs, Wald's test, DESeq2, adjusted p-value < 0.05, Fig. 6a, b), while no significant differences in the taxonomic composition or total relative sum abundance of ARGs and MGEs were detected. The enriched genes included genes conferring resistance to aminoglycoside, sulphonamide, and tetracycline as well as integrases, plasmid markers, and transposases (Fig. 6a, b, negative binomial GLMs, Wald's test, DESeq2, adjusted p-value < 0.05). The results suggest that early termination of breastfeeding might have negative health effects for infants due to an increased resistance potential of the gut microbiota against certain antibiotics, and, thus, increasing the likelihood of selecting for antibiotic resistant opportunistic pathogens able to cause infections under the right circumstances.

Antibiotic treatment of mothers during delivery (IAP) had a modest but statistically significant effect on the composition of the gut microbiota of 1-month-old infants (ADONIS, $R^2 = 0.13$, $p = 0.048$), but not on the resistome or MGE composition. The abundance of horizontally transferrable ARGs and MGEs was significantly increased in the IAP group compared to the control group still 6 months after the antibiotic prophylaxis (negative binomial GLMs, $p = 5.34e{-}03$ and $p = 1.34e{-}03$, respectively). No significant changes were observed in the abundance of ARGs in the infants between the IAP and control groups or in the abundance and composition of the microbiome, ARGs and MGEs in probiotic and placebo groups. Infants in the IAP group also had a larger number of enriched ARGs and MGEs than infants in the control group (negative binomial GLMs, Wald's test, DESeq2, adjusted p-value < 0.05, Fig. 6c–f). The effect of IAP on the infants

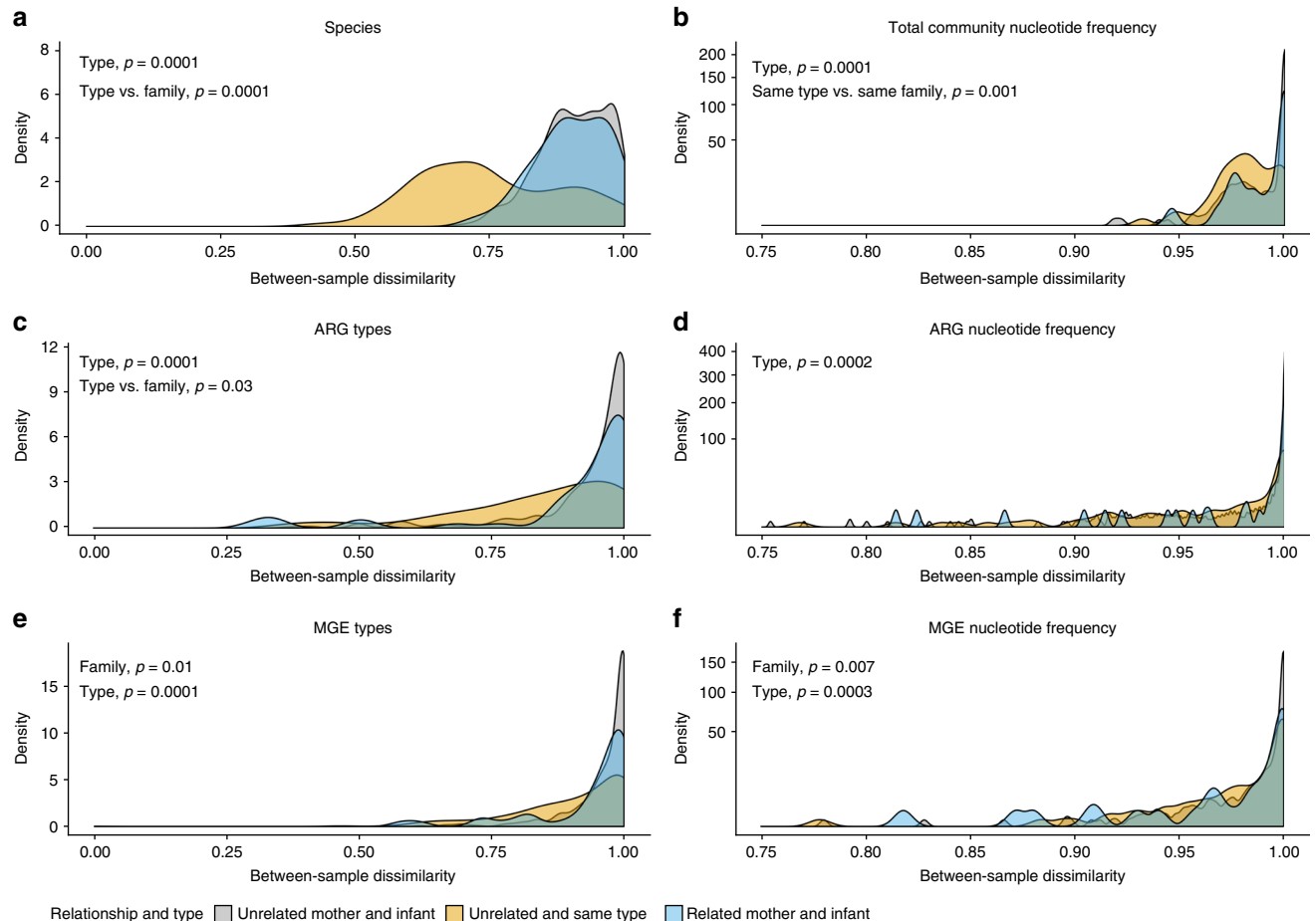

**Fig. 5** Dissimilarities of infant gut and breast milk microbiota, resistomes, and MGEs. **a** Dissimilarity of microbial communities using Metaxa2[45] taxonomy profiles based on 16S rRNA genes between breast milk and infant feces. **b** Dissimilarity of microbial communities in infants and breast milk using DNA sequence profiles calculated based on kmers. **c** Dissimilarity of resistomes of breast milk and infant's feces. **d** Similarity of resistomes between breast milk and mother's feces. **e** Dissimilarity of MGEs between breast milk and infants' feces. **f** Dissimilarity of MGEs between breast milk and mothers' feces. Dissimilarities between related and unrelated infant-mother pairs were compared. Notion type indicates that breast milk and feces are significantly different from each other, family indicates that mother's breast milk is more similar to related infant's feces than to unrelated infant's feces, type vs. family indicates that breast milk samples are more similar to each other than to feces samples from the infant from the same family, family vs. type indicates that breast milk and feces samples are more similar to samples from family members than to a sample of the same type. Significance of differences was tested using ANOVA between the similarity indexes in the comparisons and p-values < 0.05 are indicated in the figures. The density plot depicts where comparisons between sample pairs are located on the dissimilarity scale. The higher the density is at a given dissimilarity, the more pairwise comparisons have the given dissimilarity value. Density of the samples is plotted on the y-axis and the x-axis depicts the between sample Jaccard dissimilarity index of species, ARGs and MGEs shared between sample types using presence–absence data or DNA sequence profiles based on kmers calculated with sourmash[56]. DNA sequence profiles provide a way to compare DNA sequence signatures of samples with each other and does not rely on species or gene annotations. The x-axis ranges from 0 (no dissimilarity) to 1 (complete dissimilarity)

persisted for 6 months, even though no significant trends for increase in the ARG or MGE loads were observed in the mothers.

**Sharing of strains and MGEs containing ARGs**. The presence of MGEs carrying ARGs shared among mother-infant pairs was investigated. All together 86 contigs containing an ARG and an MGE were assembled from the samples (Supplementary Data 5). Of those contigs, 26 contigs with the same arrangement of genes in the contigs were shared between a related mother-infant pair the most common contig being the Tn916 transposon with a *tetM* resistance gene. This suggests that the MGEs containing ARGs are shared between family members. Nearly all of the mobile ARG contig types were found in more than one infant-mother pair, indicating that the MGE types which could be assembled from the dataset are common in the gut and that the identified

types are widely spread. However, the analysis was done based on gene annotations only, and cannot distinguish between nucleotide level variants of the genetic elements due to the technical challenge of short read assembly of low abundance genes in varying genetic contexts, such as ARGs from diverse environments[33–35].

We studied whether mothers and infants shared strains of the species, which were linked to antibiotic resistance using Strainphlan[50]. *E. coli* was the only species which shared strains between the guts of infants and their mothers and the sharing was observed only in one mother-infant pair. This suggests that infants do not generally acquire the *E. coli* strains from their mother or that the maternal strains come from other body sites. The investigation of contigs revealed that several of the assembled mobile ARG containing contigs had their best taxonomic hit to *Escherichia* or *Klebsiella* (Supplementary Data 5), indicating that many ARGs carried by *E. coli* in the infant gut are mobile. It has

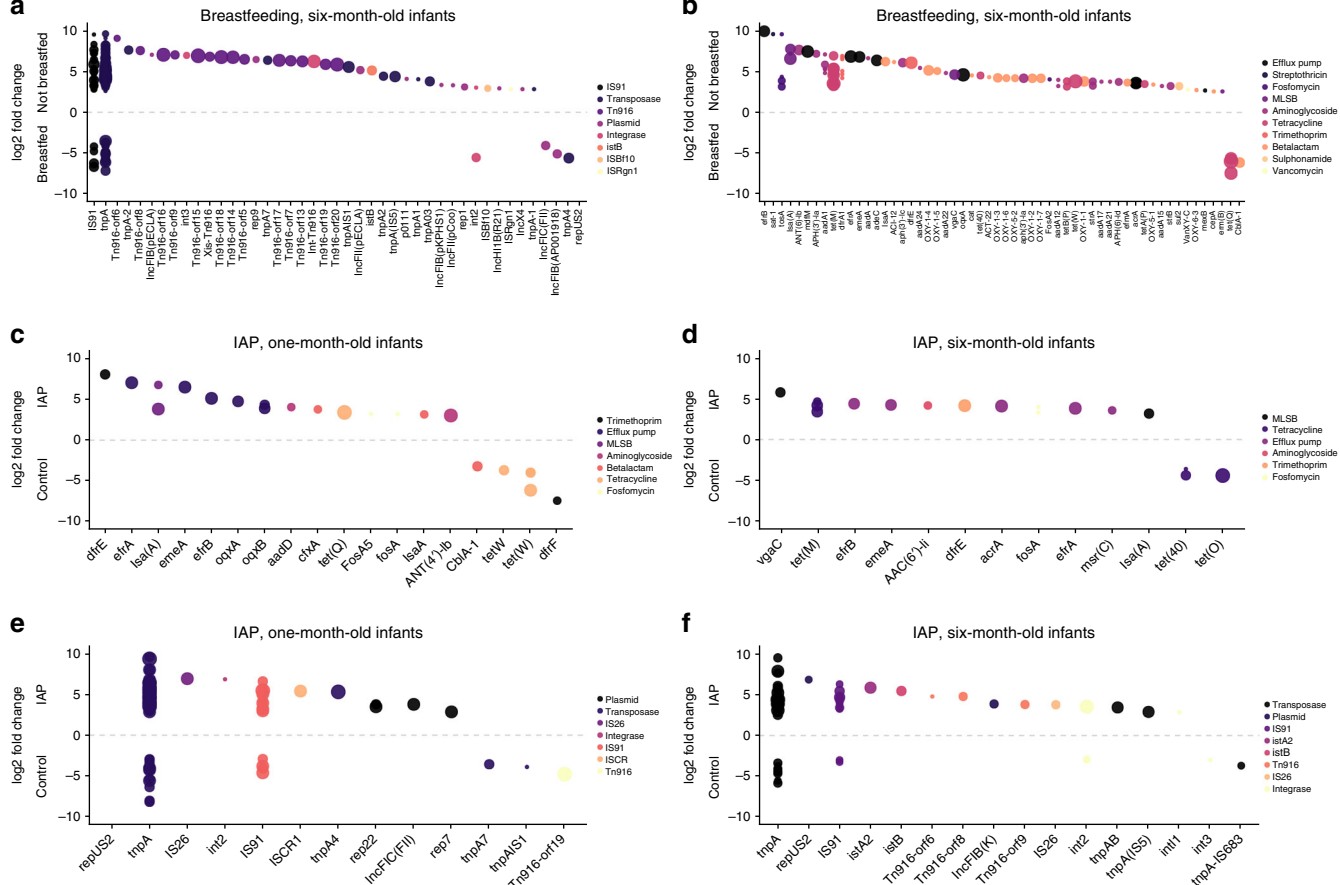

**Fig. 6** Differentially abundant ARGs and MGEs in 6-month-old infants. **a** MGEs differentially abundant due to breastfeeding in 6-month-old infants compared to non-breastfed infants. **b** ARGs differentially abundant in breastfed infants at 6 months, **c** ARGs differentially abundant in 1-month-old infants due to IAP. **d** ARGs differentially abundant in 6-month-old infants due to IAP. **e** MGEs differentially abundant due to IAP in 1-month-old infants. **f** MGEs differentially abundant due to IAP in 6-month-old infants. Genes that have negative fold changes are more abundant in the non-breastfed and IAP groups. Sizes depict the number of samples each gene was found in (n = 1–10), color represents ARG or MGE class. The y-axis shows log2 fold changes and the x-axis denotes gene names

been previously observed that Enterobacteriaceae in infants born in hospitals are likely to be acquired from the hospital environment[22,36], which might partly explain the observed correlation of *E. coli* with high ARG abundance in the infant gut.

## Discussion

Our study provides insight into the assembly of the infant resistome and MGEs during the first 6 months of life. Infants shared gut resistomes and MGEs with their own mother's gut and breast milk microbiota. Infants had higher relative abundances of both ARGs and MGEs than adults. The high abundance of MGEs may cause health risks as it is likely that resistance genes associated with MGEs are more likely to be transferred from commensal bacteria to pathogens[37]. This inherent dynamic nature of the mobilome was also observed in our study as the mobilome structure was not correlated with individual taxa and was shared between mother-infant pairs in both breast milk and feces despite the vast differences in their microbiota composition. The observed increased similarity of the resistance genes and the MGEs between infants and their mothers compared to unrelated mothers can partly explain why infants have high resistance gene loads before having antibiotic treatment as the history of antibiotic use or acquired antibiotic resistant strains of the mother could have an effect on the developing infant gut resistome

through transfer of resistant bacteria. However, it seems that some of the resistance genes could be transferred from other maternal body sites besides gut or breast milk or from non-maternal sources, which we did not investigate. Intrapartum antibiotic prophylaxis of mothers was linked to enrichment of resistance genes and mobile genetic elements in the infant gut still 6 months after the treatment, showing that antibiotic consumption of mothers can potentially have significant and persistent effects on the infant resistome and mobilome at least when the antibiotic has been administered intravenously during delivery. Some of the effects of IAP might also be passed on to infants by transmission of strains from other maternal sources. For example, the initial largescale colonization of the infant gut begins during labor, and thus, changes in the vaginal microbiota could have a significant impact on the gut microbiota of infants.

The infant gut resistomes were strongly shaped by phylogeny with *E. coli* and Gammaproteobacteria being highly positively and *Bifidobacterium* negatively correlated with resistance load. We observed enrichment of resistance and MGE-associated genes in infants who were breastfed for less than 6 months despite that breast milk, like skin[38], had high relative abundances of resistance genes. Our results indicate that breastfeeding for at least 6 months, which is known to have many health benefits[32], reduce the abundance of Gammaproteobacteria and conversely increase

the abundance of Bifidobacteria[14], could thus also be promoted for the reason that it potentially reduces the antibiotic resistance gene load in the infant gut. Based on our study and previous work[8,11] it seems that infants inherit the legacy of past antibiotic use, as they carry high loads of antibiotic resistant bacteria acquired from their mothers and the environment even prior to being exposed to antibiotics themselves. However, since phylogeny was the major determinant of the resistome and most ARGs seemed to be carried by a limited number of taxa in the infant gut, actions that shape the overall microbiota composition of the infant gut, such as breastfeeding or antibiotics, likely have a strong impact on the resistance gene load in infancy.

## Methods

**Study subjects and sample collection.** The 16 mother-infant pairs in this study were selected from 241 pairs participating in a parallel, double-blind placebo-controlled trial on the effects of probiotics in the development of eczema[39]. The study was found ethically acceptable by the Hospital District of Southwest Finland Ethical committee and informed consent was obtained from the families participating in the study. Only infants born vaginally at full term (gestational age ≥37 weeks) and their mothers were included in this study. Availability of fecal samples served as an additional criterion for inclusion. Infants exposed to antibiotic treatment during the neonatal period or the first 6 months of life were excluded from the study. Intrapartum antibiotic prophylaxis (IAP) was administered to eight mothers due to positive group B Streptococcus test ($n = 6$) or premature rupture of membranes ($n = 2$). The antibiotics used were penicillin G ($n = 7$) or Cephalothin ($n = 1$). Ten of the 16 mothers received probiotic supplements containing either Bifidobacterium longum BB536 and Lactobacillus paracasei ST11 or Lactobacillus rhamnosus LGG 2 months before and after delivery. Data on the duration of total breastfeeding was collected. Metadata of the infants and mothers is provided in Supplementary Data 1.

Fecal samples were obtained from the mothers at 32 weeks of gestation and 1-month postpartum and from the infants at the ages of 1 and 6 months. The fecal samples were collected at home and refrigerated immediately for up to 24 h after which they were frozen and stored in −80 °C until DNA extraction. Colostrum samples were collected during the first week postpartum and breast milk samples 1 month after delivery. Before sample collection, the breast was cleaned with soap and water, and breast milk was collected manually, discarding the first drops, with a sterile milk collection unit. Mothers were given written instructions for standardized collection of samples and the samples were frozen and stored at −20 °C at home and subsequently at −80 °C at the laboratory until DNA extraction.

**DNA extraction and sequencing.** DNA was extracted from fecal and milk samples using King Fisher (ThermoFisher) and InviMag® Stool DNA Kit (Stratec). Stool samples with visible coloration after DNA extraction were purified using DNeasy PowerClean Pro CleanUp Kit (Qiagen). DNA concentrations were measured using Qubit (ThermoFisher) with the Qubit dsDNA HS Assay Kit (ThermoFisher). DNA purity was determined with Nanodrop (ThermoFisher) by measuring 260/280 and 260/230 absorbance ratios. Sequencing libraries were made with Nextera XT DNA Library Preparation Kit (Illumina) according to manufacturer's instructions. Paired-end sequencing was done using NextSeq 500 (Illumina).

**Metagenomic analysis.** FastQC[40] was used to analyse the quality of the metagenomic reads. Read lengths were 170 and 132 bp for the forward and reverse reads, respectively (Supplementary Data 2 and 3). Mean PHRED of the score was 35. No quality trimming prior to further analysis was done as the quality of the reads was high.

Host DNA was removed using paired-end mapping with Bowtie2[41] version 2.3.3 against a human reference genome GRCh38.p9 GCF_000001405.35 and unmapped paired-end reads were filtered using SAMtools[42,43] version 1.4 with parameters view -b -f 12 -F 256. BEDTools[44] version 2.26.0 was used to convert the bam files to fastq files containing the non-human paired-end reads. Adapters were removed from the non-human reads with cutadapt[45] version 1.10 with default parameters. Mean sequencing depth of the non-human DNA was 1.9 Gbp for fecal samples (Supplementary Data 2) and 160 Mbp for milk samples (Supplementary Data 3). Host DNA content in the breast milk samples varied between 78 and 96%.

Species level community profiling based on marker genes was done using Metaphlan2[46] version 2.6.0 by running the metaphlan2.py command with the—input_type fastq—nproc 5 options. Merged abundance table was created using the Metaphlan2 utils script. The merged abundance table was edited to only include taxa which were identified to species level. Additional community profiling was done using Metaxa2[47] version 2.1 for 16S rRNA read extraction in paired-end mode. The 16S rRNA sequence reads were classified using mothur's[48] (version 1.39.5) classify.seqs command with SILVA[49] v. 123 as the reference database, with the cutoff = 60, probs = F and processors = 8 parameters. A custom Unix script was used to create an OTU table based on the classifications. Strain-level profiling and strain tracking analysis as done using Strainphlan[50] and the extract_markers.

py command to extract markers for Escherichia coli, Klebsiella pneumoniae and all Streptococcus and Staphylococcus species. The strainphlan.py command with—relaxed_parameters2 option was used to create taxonomic trees of the strains in all samples.

Resistome and mobilome were characterized by mapping metagenomic reads to a comprehensive non-redundant database of more than 2700 mobile genetic element and 3100 antibiotic resistance genes including Comprehensive Antibiotic Resistance Database protein homolog model version 1.1.2 (CARD)[51] and ResFinder version 2.1[52]. Some analyses were performed with only the Resfinder database, which has only acquired or mobile ARGs. A custom MGE database was created by fetching CDS for genes that were annotated as IS*, ISCR*, intI1, int2, istA*, istB*, qacEdelta, tniA*, tniB*, tnpA*, or Tn916 transposon ORFs or genes in the NCBI[53] nucleotide database and PlasmidFinder database[54]. Redundancy in the databases was removed with VSEARCH[55] -usearch_global command with—id 99 option. In total, the MGE database consists of genes with 278 different gene name annotations and more than 2000 unique sequences excluding the sequences from PlasmidFinder database. The custom MGE database is available from https://github.com/KatariinaParnanen/MobileGeneticElementDatabase.

Bowtie2[41] mapping was done with options -D 20 -R 3 -N 1 -L 20 -i S,1,0.50 and was used to map reads the ARG and MGE databases. SAMtools[42,43] was used to filter and count reads and if both reads mapped to the same gene the read was counted as one match and if the reads mapped to different genes, both were counted as hits to the respective gene.

DNA sequence profiles based on kmers were produced from the metagenomic reads using sourmash[56] with option -kmer 31. Subsets of reads mapping to ARGs and MGEs were used to produce the profiles for ARGs and MGEs.

**Preliminary data processing for statistical analysis.** Statistical analysis was done in R[57] version 3.4.1. Metaphlan2[44], Metaxa2[45] and ARG and MGE mapping results, taxonomy and annotation tables and metadata files were compiled into individual data objects in phyloseq[58] version 1.20.0. All custom R codes are available in Supplementary Software file. The ARG and MGE Bowtie2 counts were normalized to the length of the respective gene. The length-normalized numbers were then further normalized to the number of bacterial 16S rRNA gene reads divided by the length of the 16S rRNA gene. The 16S rRNA gene was chosen for normalization instead of library size to account for variation in non-bacterial DNA content in the samples. The normalized values were used in all the following analyses. All figures if not indicated otherwise were drawn with ggplot2[59] version 2.2.1.

**Ordination analysis.** Principal coordinate analyses of the taxonomic profiles ARGs and MGEs was done on presence–absence and relative abundance data using cmdscale command in R. Distances between samples were calculated using Horn-Morisita[60] similarity index obtained with the vegdist command from vegan package[61] version 2.4–3. Permutational multivariate analysis of variance (PERMANOVA) between different groups was done with adonis in vegan[61] with similarity index using 9999 permutations and the resulting p-values were corrected with Benjamini & Hochberg procedure for multiple testing using the p.adjust command in R.

**Differences in diversities and abundances of ARGs and MGEs.** Shannon and Simpson diversities and Chao1 estimate of richness of the taxonomic profiles, ARGs and MGEs were calculated using vegan[61]. Statistical significances of the differences between the diversities in sample types (infant 1 M, infant 6 M, mother 32 GW, mother 1 M, colostrum, and breast milk 1 M) and in IAP versus control, breastfeeding until 6 months and breastfeeding for less than 6 months were calculated using analysis of variance (ANOVA) and the resulting p-values were corrected using Tukey's post hoc test[62,63] in base R.

Statistical significances of differences in ARG and MGE total sum relative abundances normalized to total 16S rRNA gene counts and length were calculated using negative binomial generalized linear models (GLMs) in the MASS package[64] version 7.3–47 and corrected using Tukey's post hoc test[62,63]. Negative binomial GLMs were used since the distribution of ARGs and MGEs did not fit normal or Poisson distribution due to overdispersion.

**Analysis of differential abundances of ARGs, MGEs, and taxa.** Analysis of which genes' relative abundances differ due to breastfeeding duration and IAP, as well as taxa which change between 1- and 6-month-old infants and infants and mothers was done using DESeq2[25] version 1.16.1 to obtain fold changes and statistical significances of the observed changes. Before analysis with DESeq2[25], gene counts were normalized to total 16S rRNA gene counts and gene lengths and then multiplied by 1e5 (5.6 times the mean 16S rRNA read count) and rounded to integers resulting in normalized values which take into account variation in the 16S rRNA gene counts in different samples. Pseudo-count of plus one was added to all values, so that zero values were changed to one, which enables comparisons between not detected and detected values (6.6 versus 1 for a detected gene versus not detected). Several different values for detected genes were used ranging from 210 and similar results were obtained in the different iterations showing that the

analysis was robust regardless of the selected values for normalization. A significance cutoff for the adjusted p-value of 0.05 was used in the analysis.

**Mantel's test and correlation of taxa with ARG and MGE abundances.** Species, ARG and MGE distance matrixes were calculated using Horn-Morisita similarity index[60] with vegdist command in vegan package[61] and compared to observe if there were correlations between taxonomy and the resistome and mobilome. Comparisons were done using Mantel's test from vegan[61] and Pearson pair-wise correlations obtained using rcorr command from the Hmisc[65] package version 4.0–3.

Negative binomial GLMs from the MASS package[64] were used to find taxa on class, genus, and species level which correlated with the ARG and MGE total sum abundances. All the taxa on the investigated taxonomic levels were correlated with the ARG or MGE total sum abundance in the different sample types (1- and 6-month-old infants and pregnant and 1 month postpartum mothers). The p-values were corrected using p.adjust command with Benjamini & Hochberg method. After finding candidate taxa, model selection was done using ANOVA and $\chi^2$-tests in base R with a significance cutoff of 0.05.

**Venn diagrams.** The number of present and shared species or gene types were calculated for the compared sample types and Venn diagrams were drawn in R using the VennDiagram package[66] using draw.pairwise.venn command.

**Similarity of gene and taxa profiles between mothers and infants.** Vegan package[61] and Jaccard dissimilarity indices using presence–absence data were used to calculate distance matrixes for mothers 1 month postpartum and 1-month-old infants, and infants at 1 and 6 months of age to determine whether the microbiomes, resistomes, and mobilomes were more similar within samples collected from the same family or from the same infant at different times. Similar analysis was also done using DNA sequence profiles of the total microbial community, ARGs and MGEs to obtain nucleotide level comparisons between samples based on kmer frequencies of the metagenomic reads (all non-human reads and reads mapping to ARGs and MGEs, respectively). The DNA sequence profiles provide means to compare the signatures samples to each other on a kmer level independently of database-based annotations. Kmer frequency signatures were produced using sourmash[56] with kmer size 31. For species comparisons the data was square root transformed. Results were presented with density plots produced using ggplot2[59]. Statistical analysis of the significances of observed differences was done by linear modeling with lm function in base R (see function lmp in Supplementary Software file). Permutations of how often a smaller p-value was achieved with randomized data as opposed to the assigned two groups compared was used to calculate the corrected p-value. 9999 permutations were used for the analysis. The effect of IAP on the frequency of sharing taxa or ARGs and MGEs was done in a similar manner as described above, but the distances were calculated between related mother-infant pairs in IAP and control groups and the effect of IAP was determined on based whether there was a significant difference in similarities due to the antibiotic treatment. Related mother-infant pairs were analyzed in a similar fashion, but the distances were calculated between samples which were taken at the same time (1 month time point) or at different times (antepartum maternal samples compared to infants at 1 and 6 months as well as 1 month postpartum maternal sample compared to 6-month-old infants).

**Code availability.** R-code for statistical analyses are available in Supplementary Software file.

## Data availability
The raw sequence files are available under NCBI Bioproject PRJNA384716. The custom MGE database is available at Github (https://github.com/KatariinaParnanen/MobileGeneticElementDatabase).

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

## Acknowledgements

The research was supported by grants from Academy of Finland (grant 1268643), the Sigrid Juselius Foundation and Turku University Hospital EVO research funding. S.R. has received research support from the Sigrid Juselius Foundation, EVO research funding of Turku University Hospital, the Academy of Finland, and the Finnish Society for Pediatric Research. S.R. received the probiotic strains from Nestle without compensation. S.S. received travel expenses from the European Food Safety Authority. E.I. received research support from the Sigrid Juselius Foundation and lecture fees and travel expenses from International Academic Meetings. R.S. received support from the Academy of Finland (grants 258439 and 283088). J.B.-P. received support from Swedish Research Council for Environment, Agricultural Sciences and Spatial Planning (FORMAS; grant 2016-00768). K.P. received funding from the MBDP doctoral program at the University of Helsinki. CSC, IT Center for Science, Finland, is acknowledged for providing the computational resources for the study.

## Author contributions

K.P. did laboratory work, bioinformatic, and statistical analysis, wrote the manuscript and participated in conceiving the study. R.S., S.S., E.I., C.L. and M.V. participated in conceiving the study. A.K. and J.H. assisted in bioinformatics and statistical analysis. J.B.-P. assisted in statistical analysis and gave input on the manuscript. D.G.J.L. gave guidance on aspects to analyze in the data and gave input on the manuscript. S.R., S.S., H.K. and E.I. provided the samples and clinical data. All authors contributed to manuscript revisions and have read and approved the final version of the manuscript.

## Additional information

**Competing interests:** The authors declare no competing interests.

