## [Peer Review File · Nature Communications]

Reviewers' comments:

Reviewer #1 (Remarks to the Author):

The understanding of the origin and transmission of antibiotic resistance genes is definitely of major scientific interest. The analytic work seems thorough, with the text being quite easy to follow. The cohort, however, is quite small, and does not represent a normal population. There are also additional issues that need to be addressed more in depth:

- a) The potential that the bias in detected ARG and MGA is due to that elements have been thoroughly studied in Enterobacteriaceae, so that it can be difficult to relate true differences for samples containing other less characterized families
- b) The sequencing depth of mothers milk samples is 1/10 of the other sample types. The potential bias due to differences in sequencing depths should be addressed more thoroughly, in addition to the potential of contaminating DNA.
- c) Matching of sequences from different samples due to the hit to the same target does not necessary mean that the sequences in the samples have the same origin. I would expect more direct sequence based comparisons

Detailed comments:

"Infant gut microbiomes had higher abundances of ARGs and genes associated with MGEs than the gut microbiomes of their mothers ($p < 0.05$) (Figure 1, Supplementary Table 2). This was apparent even though the infants had not been exposed to antibiotics during their life."

Comment: Half of the mothers received intrapartum antibiotic prophylaxis in order to prevent mother-child transmission of pathogens. In order to address effects of ARG and MGE transmission without antibiotic exposure, this should have been analyzed separately for the mothers without antibiotic treatment. A direct linkage between ARG and MGE should be provided (if they are identified on the same contigs).

Figure 1 is difficult to understand – it is too busy. I would suggest a figure only highlighting the significant findings, then present the rest as a supplement. Why were Shannon and not Simpson diversity index used, since Shannon is very sensitive to low abundant observations. Due to the major differences in sequencing depth between fecal and mothers milk samples, I think that would affect the low abundant samples. I also think that the authors should state why the ratio between 16S rRNA gene and ARG/MGE is not affected by sequencing depth.

"Breast milk harbored a microbial community, resistome and MGE profile distinct from the gut (ADONIS, adjusted p -value < 0.05 , Figure 2, Supplementary table 2). "

Comment: I think comparisons also should be done for presence/absence data, as for Figure S3 for the different categories. This would enable direct statistical evaluation of the mother/child associations. In my view presence/absence data would be the most relevant for addressing transmission.

"Despite the fact that breast milk had similar total abundances of ARGs to fecal samples, the abundant taxa in breast milk samples were negatively correlated with the total relative ARG abundances in infants (adjusted p -value < 0.05 , Supplementary table 3). Bifidobacterium had the strongest negative correlation with ARG abundance in infants (Supplementary table 3)."

Comment: I think this is really interesting. However, Bifidobacteria does not seem abundant in mothers milk. What is the actual abundance of Bifidobacteria?

"Infants shared 40% of their gut ARGs and 37% of their MGEs with the maternal gut microbiota and correspondingly, 20% and 12% with breast milk (Supplement figure S3)."

Comment: I think this statement is unclear. Would it mean same type, or identical?

“Figure 4: Density plots of dissimilarities in gut microbiota composition, antibiotic resistance genes (ARGs) and mobile genetic elements (MGEs) between one month postpartum mothers and one-month-old infants and between infants at one and six months”

Comment: I had a hard time trying to understand the plot. I would suggest a graphical representation, highlighting what is described in the text. Is the comparison based on the matching towards the same elements in the database, if so – how can you actually know they are the same?

“Previous studies have not been able to pinpoint that infants share significantly more ARGs with their mothers than with unrelated adults possibly due to small set of subjects or methodological challenges^{8,12}. Our results suggest that maternal ARGs and MGEs have significant effects on the resistome and MGE composition of the infant gut and that there likely is vertical transfer of some of these genes between mothers and infants.”

Comment: Could the differences be related to intrapartum antibiotic prophylaxis in 50% of the study population?

“Figure 5: Density plots of dissimilarities in microbiota composition, antibiotic resistance genes (ARGs) and mobile genetic elements (MGEs) between breast milk and one-month-old infant gut and between breast milk and maternal gut”

Comment: Same comment as for Fig. 4

“Antibiotic treatment of mothers during delivery (IAP) had a modest but statistically significant effect on the gut microbiota of one-month-old infants (ADONIS, $R^2=0.13$, $p=0.048$). The abundance of horizontally transferrable ARGs and MGEs was significantly increased in the IAP group compared to the control group still six months after the antibiotic prophylaxis (Negative binomial GLMs, $p=5.34e-03$ and $p=1.34e-03$, respectively). Infants in the IAP group also had a larger number of enriched ARGs and MGEs than infants in the control group (DESeq2, adjusted $pvalue < 0.05$, Figure 4). The effect of IAP on the infants persisted for six months, even though no significant trends for increase in the ARG or MGE loads were observed in the mothers.”

Comment: From the metadata IAP seems like the main factor. I would have expected stratification of the results based on AIP earlier in the results description. Is IAP mentioned in Fig 4?

“We studied if mothers and infants shared strains of the species which were linked to antibiotic resistance using Strainphlan²¹. *E. coli* was the only species which shared strains between the guts of infants and their mothers and the sharing was observed only in one mother-infant pair. This suggests that infants do not generally acquire the *E. coli* strains from their mother. The investigation of contigs revealed that several of the assembled mobile ARG containing contigs had their best taxonomic hit to *Escherichia* or *Klebsiella* (Supplementary table S4), indicating that many ARGs carried by *E. coli* in the infant gut are mobile. It has been previously observed that *Enterobacteriaceae* in infants born in hospitals are likely to be acquired from the hospital environment^{21,30}, which might partly explain the observed correlation of *E. coli* with high ARG abundance in the infant gut.”

Comment: For the other analyses how could you know that there are sharing of the same elements without exact sequence matching. Matching of two sequences towards the same elements in the database would not mean that the two sequences are identical. I generally find it unclear what the basis for the matches described through the manuscript actually reflect.

“Our study provides insight into the assembly of the infant resistome and MGEs during the first six

months of life. Infants shared their resistomes and MGEs with their own mothers mediated via vertical transfer from mother's gut and breast milk microbiota."

Comment: I don't think vertical transmission of MGE has been shown.

"Based on our study and previous work^{8,11} it seems that infants inherit the legacy of past antibiotic use as they carry high loads of antibiotic resistant bacteria acquired from their mothers and the environment even prior to being exposed to antibiotics themselves."

Comment: I would guess IAP would matter.

"However, actions that shape the overall microbiota composition of the infant gut, such as breastfeeding or antibiotics, likely have a strong impact on the resistance gene load, as phylogeny was the major determinant of the resistome and most ARGs seem to be carried by a limited number of taxa in the infant gut."

Comment: How would IAP affect the vaginal resistome and ARG content, and what could the potential role of the vaginal microbiota be for the mother to child transmission of ARG?

Reviewer #2 (Remarks to the Author):

I prefer an anonymized report. Call me old school, but it is my preference.

This is a very exciting and interesting study regarding the source origin of both ARGs and MGEs in the infant gut microbiome and the breast milk microbiome.

While there are multiple exciting elements to this study, there are also some fundamental key limitations which restrict both interpretation and findings.

Exciting aspects

1. This study is inherently novel in its aims and goal of looking for the source of the neonatal/infant resistome. Namely, they sought to characterize both ARGs and MGEs in the maternal gut and breast milk microbiomes, then correlate these with the neonate/infant stool patterns. They study 16 mother-infant pairs over a span of up to 8 months, with neonates sampled at 1 month and infants sampled at 6 months. Mothers were sampled at 32 weeks gestation and one month post partum. Thus, there are unpaired maternal and infant samples at 32 weeks (moms only collected, fetal collection clearly not possible) and again at 6 months (infant only, no maternal breast milk nor stool).

Thus the sample collection is both exciting and an inherent strength but also a significant limitation. This is further discussed below. As such limitations are inherent and evident, the abstract and last paragraph of introduction must be rewritten as they overstate the design and are not reflective of what was actually collected and done.

2. The methodology (if one ignores the study design limitations and concerns) is very sound and I have no concerns about the metagenomics analysis and pipelines used per se. They are very standard and appropriate. As a minor point, measuring significance in their beta diversity by PERMANOVA or Adonis would be good--it is alluded to in the text but not placed on the legend nor in Figure 2.

The exciting part is the deep sequencing of the microbiome in the human milk and the tools they

used to do so. This is a relatively novel application of existing tools, and is good to see done.

3. The degree to which the milk and infant gut harbor ARGs and MGEs is a key contribution to the literature. Although not novel (see cited Gibson paper) this is well done and from a cohort of term, vaginally delivered infants.

That said, the lack of other maternal and infant body sites, including placenta and oral and skin, is a real missed opportunity and limits all source and sink discussions. This is further confounded by the fact that the maternal gut microbiome is known to change in pregnancy, but when it resumed pre-pregnant community structure and function is not yet known. Assuming a maternal composition at 32 weeks and 1 month will be representative of 6 months is not appropriate.

Limitations

1. Sample collection is actually unpaired and quite limited. Trying to trace the source of neonatal/infant ARGs and MGEs with nothing but 1 week and 1 month maternal breast milk and 32 week and 1 month stool is inherently problematic and the fundamental limitation to the study. Since the infant stool is collected at one and six months, there is actually only a single temporal coincident time point (1 month) and while valuable is still a significant limitation.

Beyond the temporal limitations, there is also concern for tracing source only from breastmilk and stool. The investigators ignore the data, notably regarding *E. coli* and strep and klebsiella, from multiple labs demonstrating a placental microbiome. They similarly neglect studies from multiple labs looking at the early onset diversity of multiple infant body sites, their differences, and sources from placenta and maternal stool, breastmilk, vagina, and oral communities. In a nutshell, making huge source assumptions looking only at unpaired maternal milk and stool is a tremendous leap. At present, the conclusions are not supported by the presented data.

3. The figures would benefit from some editing. The axis labels are not particularly helpful and presenting box and whiskers of Shannon diversity without other alpha measures (Chao 1, beta diversity) is a key limitation. The data in Figure 1 is really not novel nor well done, and do not heighten enthusiasm for the study and its findings.

4. Figure 4 is inherently flawed for all the reasons discussed above. Running BC dissimilarity on actually non-paired samples and without other body sites is disingenuous and the conclusions cannot be considered valid as they were never actually tested. The same applies to Figure 5 and Figure 6.

5. A final real missed opportunity and neglected key point is the number of subjects on probiotic supplements. This is crucially important, since this same team of investigators has previously published data suggesting maternal to infant transfer of probiotics occurs in utero and during lactation. However, there is literally no demonstration of this spp and strains in their metagenomic analysis in this paper. This renders the formal possibility that their unaccounted for ARGs and MGEs could actually be arising from these probiotics. This must be tested, since 10/16 mothers were on them.

Reviewer #3 (Remarks to the Author):

The major findings of this paper relate to how the microbiomes, resistomes, and mobilomes of infants are shaped by their mother and by breast milk. There are several novel findings in the paper and this will be of interest to others in the field. The statistics appear to be well chosen and

applied, convincing the reader that the claims are valid.

However, the manuscript is very wordy and can be shortened. Many comments in the text seem to have no greater purpose or implication. For example:

"Transposon Tn916 associated genes were found in all sample types but were especially common in breast milk and in one-month-old infants."

Why was that specific MGE was chosen for comment? There are other MGE that are also found in all sample types but more common in some than others (as seen in the Figure).

Some other comments:

Breast milk, as the primary source of nutrition during the first months, shapes the infant gut microbiota^{14,15}, but its role in contributing to the resistome is virtually unexplored.

There have been studies, (e.g. <http://msystems.asm.org/content/3/1/e00123-17>) that have explored how breast milk vs formula feeding affects the resistome.

Figure 1- please make the letters referring to each part of the figure panel larger. Also, to avoid confusion, when referring to parts of the panel in the text, include the letter -- i.e. use Figure 1E rather than Figure 1 in

"The relative abundances of ARGs were similar in breast milk and fecal samples from six-month-old infants and mothers ($p > 0.05$, Figure 1, Supplementary Table 2)."

"The results suggest that early termination of breastfeeding might have negative health effects for infants due to an increased resistance potential of the gut microbiota against certain antibiotics." Authors may want to comment on why this is a negative effect. In the introduction, the authors discuss the negative aspects of infectious pathogens carrying ARGs, but has it ever been shown that higher ARG in the microbiota leads to negative health effects?

Reviewer 1

The understanding of the origin and transmission of antibiotic resistance genes is definitely of major scientific interest. The analytic work seems thorough, with the text being quite easy to follow. The cohort, however, is quite small, and does not represent a normal population. There are also additional issues that need to be addressed more in depth:

We assume that the remark regarding normal population is related to the IAP treatment received by mothers. This is answered in below in the answer to “*Comment: Half of the mothers received intrapartum antibiotic prophylaxis in order to prevent mother-child transmission of pathogens. In order to address effects of ARG and MGE transmission without antibiotic exposure, this should have been analyzed separately for the mothers without antibiotic treatment.*”, but briefly we did not observe significant differences due to IAP treatment in the transmission of ARGs (8 mothers with IAP and 8 without) and, thus, trust that our conclusions of ARG transmission are valid also for mothers and infants without IAP treatment.

We agree that our cohort was quite small but would like to argue that the cohort size was large enough to test our hypotheses, i.e. whether infants are more similar to their own mothers than to unrelated mothers. The transmission hypothesis was tested with pairwise comparisons and obtaining a result which shows more similarity between related mothers and infants than unrelated mothers by chance is unlikely. There might be some uncertainty in estimating the effects of breastfeeding and IAP on the ARG and MGE abundance in the infant gut due to the small cohort size. However, bifidobacterial abundance is known to increase as a consequence of breastfeeding and correspondingly the abundance of enterobacteria to decrease based on several studies. *Enterobacteriaceae* are also commonly resistant to antibiotics, and thus our observation that breastfeeding might decrease the abundance of certain ARGs and MGEs in the infant gut is also supported by literature. Correspondingly, we trust that our observation that IAP might increase ARGs and MGEs in the infant gut can be expected since antibiotic treatment can select for antibiotic resistant bacteria.

a) The potential that the bias in detected ARG and MGA is due to that elements have been thoroughly studied in Enterobacteriaceae, so that it can be difficult to relate true differences for samples containing other less characterized families

We agree with Reviewer 1 that there are inherent limitations in analysis of ARGs and MGEs from metagenomes based on databases. The databases have a bias towards *Enterobacteriaceae* and other clinically relevant species, which can affect the results. We miss uncharacterized genes from other less studied bacterial species and thus, the differences in the samples reflect only known ARGs and MGEs. That said, we are interested mostly in clinically relevant species and genes, which cause the most risk to infant health, and thus, we believe that this bias in the databases is tolerable. Functional metagenomics would allow for database independent studying of also unknown ARGs, but this is a highly resource intensive technology, and not very quantitative nor as sensitive as metagenomics due to lower throughput. Also, it cannot easily distinguish between mobile and non-mobile resistance factors, and it would not work to identify MGEs.

b) The sequencing depth of mothers milk samples is 1/10 of the other sample types. The potential bias due to differences in sequencing depths should be addressed more thoroughly, in addition to the potential of contaminating DNA.

The majority of the results in the manuscript are from gut samples which have very similar library sizes. We have added text addressing the fact that there is uncertainty in the abundance of ARGs and MGEs in breast milk due to lower sequencing depth. However, in principle there is an equal chance of sampling 16S rRNA reads and ARG reads at random, and the sampling frequency of each is dependent on their respective abundances in the samples. We addressed the differences in sequencing depth by choosing statistical methods that are robust for differing sample sizes such as negative binomial GLMs and Horn-Morisita dissimilarity index when we compare breast milk to feces. When we assess transmission from breast milk to infant feces, the bias is equal for all mother-infant pairs, and thus did not infer with being able to detect transmission of MGEs. However, we might have missed transmission of ARGs and species, due to this, which has been discussed in the main manuscript.

Contamination is difficult to assess in metagenomes, since so far, including negative controls in metagenome libraries is technically too challenging to do at a reasonable cost. However, since the number of bacterial sequences is low due to the dominance of human DNA in the milk samples, only the most abundant species and genes can be detected, which lowers the chance of having major contamination driving the results as it would require a substantial number of contaminating bacteria in the sample. Milk itself has relatively high numbers of bacteria, approximately 1000-10000 CFUs/ml, and thus, does not pose the same problems with contamination as samples with very low bacterial biomass. Furthermore, contamination is likely less of an issue when the DNA is not pre-amplified prior to sequencing like in 16S rRNA amplicon studies. However, it is likely that many of the bacteria can be the same as on the skin of the mother, since they are often shared with milk. Nevertheless, the infant would consume the bacteria of skin during breastfeeding.

c) Matching of sequences from different samples due to the hit to the same target does not necessarily mean that the sequences in the samples have the same origin. I would expect more direct sequence based comparisons

We thank Reviewer 1 for the comment. Unfortunately, metagenomics and especially assembly of short metagenomic reads have known challenges when it comes to studying assembled ARG and MGE gene variants. The three main issues in assembly of ARGs and MGEs are 1) assembly algorithms perform poorly when genes are found in different contexts and the assembly breaks when alternative contexts occur, which is often the case for ARGs 2) assemblies are chimeric and represent a consensus of all the variants in the metagenome library 3) sequencing depth is often a major limiting factor even when sequencing libraries are deep, because ARGs and MGEs are relatively very low in abundance, in the range of 10^{-3} to 10^{-5} ARG reads per 16S rRNA reads. Thus, even current state of the art methods in metagenomics and assembly do not allow for this type of analysis of ARGs and MGEs on a “gene to gene” basis and resorting to fingerprint type methods is often the only possibility. We have added citations relating to the technical difficulties in assembly in the section “Sharing of strains and mobile genetic elements containing antibiotic resistance genes between infants and mothers”.

We explored different options in doing sequence level comparisons between the ARGs and MGEs of infants and their mothers and discovered a tool which enabled us to produce unique DNA sequence signatures based on reads of the pools of ARGs and MGEs for each sample. DNA sequence profile analysis was done using kmer frequency calculations of the reads of the total community and reads mapping to ARGs and MGEs using a bioinformatic program called sourmash, which calculates DNA sequence signatures based on kmer frequencies for each sample. Distances between the DNA signature profiles of the samples were computed using Jaccard similarities.

We have added the DNA sequence profile -based analysis to address the origin of the genes on a nucleotide profile level rather than based on gene type annotations to the main manuscript in Figures 4 and 5. The results obtained using the DNA-sequence profiles strongly supported the results obtained using gene types. Infants were observed to be significantly more similar to their own mothers than unrelated mothers and additionally, the observed similarities of individuals at different ages or between breast milk and maternal gut were affirmed by using the DNA signature profiles. The associations were even more significant than when using gene types in several of the occasions.

Detailed comments:

“Infant gut microbiomes had higher abundances of ARGs and genes associated with MGEs than the gut microbiomes of their mothers ($p < 0.05$) (Figure 1, Supplementary Table 2). This was apparent even though the infants had not been exposed to antibiotics during their life.”

Comment: Half of the mothers received intrapartum antibiotic prophylaxis in order to prevent mother-child transmission of pathogens. In order to address effects of ARG and MGE transmission without antibiotic exposure, this should have been analyzed separately for the mothers without antibiotic treatment.

This is a very relevant point of the reviewer. We have done such analysis on the relative abundance of ARGs and MGEs in the IAP versus control group, which is described in the section "Cessation of breastfeeding before six months and intrapartum antibiotic prophylaxis increase specific ARGs and MGEs in infants" in the second paragraph where we address the effects of IAP and breastfeeding on the MGEs and resistome. We did not observe any statistically significant differences in the relative abundance of ARGs in infants in the IAP vs non-IAP groups. However, we did observe significant differences in the abundance of transferrable ARGs and MGEs in six-month-old infants between the two groups as mentioned in the section discussing the effects of IAP.

We have replied to the comment related to transmission in more detail below in the response related to comment the "Could the differences be related to intrapartum antibiotic prophylaxis in 50% of the study population?", but briefly we did not detect differences in sharing ARGs and MGEs between IAP and non-IAP groups (new Figure S6).

Detailed comments:

“Infant gut microbiomes had higher abundances of ARGs and genes associated with MGEs than the gut microbiomes of their mothers ($p < 0.05$) (Figure 1, Supplementary Table 2). This was apparent even though the infants had not been exposed to antibiotics during their life.”

A direct linkage between ARG and MGE should be provided (if they are identified on the same contigs).

We were not able to produce assemblies of the metagenome libraries that were suitable for assessing direct linkages between ARGs and MGEs comprehensively, since only 86 of the assembled contigs contained both ARGs and MGEs and the assembly was done on 64 samples, which gives on average on less than two contigs containing mobile ARGs per sample. Difficulty in assembling mobile genetic elements from diverse environments is a common problem and currently no assembly algorithm performs well under the challenge of assembling the mosaic structure of mobile elements containing ARGs and the assemblies can only produce short contigs which are usually chimeric consensus-based sequences. Secondly the abundance of ARGs is much lower than the abundance of housekeeping genes, and despite the relatively high sequencing depth, the depth is not adequate to produce comprehensive assemblies of ARG containing genetic regions. We have added explanations of the technical difficulties in assembly in the section “Sharing of strains and mobile genetic elements containing antibiotic resistance genes between infants and mothers”.

We assume that the sentence in the manuscript cited by Reviewer 1 has been unclear as we were referring to MGEs and genes which are not strictly classified as MGEs, such as integrons. We have modified the text to be clearer and replaced the “genes associated with MGEs” phrase with “MGEs”, since our intent in this sentence was not to discuss genes directly linked on same contigs as this was technically too demanding.

Figure 1 is difficult to understand – it is too busy. I would suggest a figure only highlighting the significant findings, then present the rest as a supplement. Why were Shannon and not Simpson diversity index used, since Shannon is very sensitive to low abundant observations. Due to the major differences in sequencing depth between fecal and mothers milk samples, I think that would affect the low abundant samples. I also think that the authors should state why the ratio between 16S rRNA gene and ARG/MGE is not affected by sequencing depth.

Figure 1 has been modified to only include the most significant findings. We reanalysed the diversities using Simpson index and added Shannon diversity results to the supplement (Supplementary figure 1) and did not observe any differences in the results. We have added discussion related to the uncertainty in the estimation of the relative abundance and diversity of ARGs and MGEs in breast milk, which has lower sequencing depth than gut in the results section. However, in theory sequencing depth should not affect the ratio between 16S rRNA gene and ARGs and MGEs since how often they are sampled should only be dependent on their abundance in the total DNA extracts.

“Breast milk harbored a microbial community, resistome and MGE profile distinct from the gut (ADONIS, adjusted p -value<0.05, Figure 2, Supplementary table 2). “

Comment: I think comparisons also should be done for presence/absence data, as for Figure S3 for the different categories. This would enable direct statistical evaluation of the mother/child associations. In my view presence/absence data would be the most relevant for addressing transmission.

We thank Reviewer 1 for the comment. We have added analysis of presence/absence data (the new Figure S2 and Supplementary table 2) and changed the analyses in Figure 4 and Figure 5 to be presence/absence based.

“Despite the fact that breast milk had similar total abundances of ARGs to fecal samples, the abundant taxa in breast milk samples were negatively correlated with the total relative ARG abundances in infants (adjusted p-value <0.05, Supplementary table 3). Bifidobacterium had the strongest negative correlation with ARG abundance in infants (Supplementary table 3).”

Comment: I think this is really interesting. However, Bifidobacteria does not seem abundant in mothers milk. What is the actual abundance of Bifidobacteria?

Their abundance is low undetectable or approximately 1% based on 16S rRNA gene amplicon sequencing results, and therefore detecting them in breast milk with current limitations in metagenome sequencing depth is difficult as the majority of sequences in milk are human-derived. The abundance of *Bifidobacterium* was too low to approximate based on the metagenomic libraries in breast milk even though it was the most abundant genus in feces. We have added the estimated abundance of *Bifidobacterium* in breast milk to the manuscript based on literary references.

“Infants shared 40% of their gut ARGs and 37% of their MGEs with the maternal gut microbiota and correspondingly, 20% and 12% with breast milk (Supplement figure S3).”

Comment: I think this statement is unclear. Would it mean same type, or identical?

We mean same type. This has been clarified in the text. We were not able to produce DNA signature comparisons for the genes on a gene to gene basis, as we have explained above in the response to the general comment c), but only able to look at the whole pool of ARGs and MGEs and calculate the distances between sample pairs. Thus, the Venn diagrams where we discuss sharing of gene types based on annotations of genes, could not be produced using DNA signatures as this would require tools which enable gene to gene comparisons.

“Figure 4: Density plots of dissimilarities in gut microbiota composition, antibiotic resistance genes (ARGs) and mobile genetic elements (MGEs) between one month postpartum mothers and one-month-old infants and between infants at one and six months”

Comment: I had a hard time trying to understand the plot. I would suggest a graphical representation, highlighting what is described in the text. Is the comparison based on the matching towards the same elements in the database, if so – how can you actually know they are the same?

Figure 4 was intended to be a graphical representation of what is described in the text. We have clarified figure, figure legends and the text in the main manuscript by modifying figures and legends and by adding the notations in the figure to the main text and specifying which panels of the figure we mean when discussing the results. The figures in the main text also now contain only

comparisons between mothers and infants or breast milk and infant feces. We hope that the new representation is clearer.

We have done the matching based on finding the same elements in the database. Thus, it was not possible to definitely say that the elements are identical using fragmentary metagenomic sequence data. The sequencing depth, despite being deep by current standards, was not enough to compare the gene variants to each other even when using unassembled reads. Therefore, we have now also included DNA profile signature comparisons for the total microbial community, ARGs and MGEs, as Reviewer 1 suggested. This by no means provides a definite proof, but still adds further credibility to our results. The technical details of the new analysis are explained in the response to the general comment c).

“Previous studies have not been able to pinpoint that infants share significantly more ARGs with their mothers than with unrelated adults possibly due to small set of subjects or methodological challenges^{8,12}. Our results suggest that maternal ARGs and MGEs have significant effects on the resistome and MGE composition of the infant gut and that there likely is vertical transfer of some of these genes between mothers and infants.”

Comment: Could the differences be related to intrapartum antibiotic prophylaxis in 50% of the study population?

We thank Reviewer 1 for the comment. We did new analysis on whether there is a difference in how similar the ARGs and MGEs profiles between mothers and their infants dependent on if the mothers received IAP antibiotics. There was no statistically significant difference in the similarities between IAP and control groups for the ARGs and MGEs. However, there was a difference in sharing of species, so that the control group's mothers and infants were more similar to each other than in the IAP group. The analysis is in the new supplementary figure Figure S6 and the results have been added to the main text.

“Figure 5: Density plots of dissimilarities in microbiota composition, antibiotic resistance genes (ARGs) and mobile genetic elements (MGEs) between breast milk and one-month-old infant gut and between breast milk and maternal gut”

Comment: Same comment as for Fig. 4

This has been addressed in the response to the previous comment.

“Antibiotic treatment of mothers during delivery (IAP) had a modest but statistically significant effect on the gut microbiota of one-month-old infants (ADONIS, $R^2=0.13$, $p=0.048$). The abundance of horizontally transferrable ARGs and MGEs was significantly increased in the IAP group compared to the control group still six months after the antibiotic prophylaxis (Negative binomial GLMs, $p=5.34e-03$ and $p=1.34e-03$, respectively). Infants in the IAP group also had a larger number of enriched ARGs and MGEs than infants in the control group (DESeq2, adjusted $pvalue < 0.05$, Figure 4). The effect of IAP on the infants persisted for six months, even though no significant trends for increase in the ARG or MGE loads were observed in the mothers.”

Comment: From the metadata IAP seems like the main factor. I would have expected stratification of the results based on AIP earlier in the results description. Is IAP mentioned in Fig 4?

We thank Reviewer 1 for the comment. We observed that mothers in the IAP group did not share significantly more MGE or ARGs gene types with their infants than mothers in the control group. This analysis has been added in the new Figure S6 and main text where we compare if there is stratification due to IAP in sharing of ARGs, MGEs and species.

“We studied if mothers and infants shared strains of the species which were linked to antibiotic resistance using Strainphlan21. E. coli was the only species which shared strains between the guts of infants and their mothers and the sharing was observed only in one mother-infant pair. This suggests that infants do not generally acquire the E. coli strains from their mother. The investigation of contigs revealed that several of the assembled mobile ARG containing contigs had their best taxonomic hit to Escherichia or Klebsiella (Supplementary table S4), indicating that many ARGs carried by E. coli in the infant gut are mobile. It has been previously observed that Enterobacteriaceae in infants born in hospitals are likely to be acquired from the hospital environment^{21,30}, which might partly explain the observed correlation of E. coli with high ARG abundance in the infant gut.”

Comment: For the other analyses how could you know that there are sharing of the same elements without exact sequence matching. Matching of two sequences towards the same elements in the database would not mean that the two sequences are identical. I generally find it unclear what the basis for the matches described through the manuscript actually reflect.

Our response to general comment c) addresses this concern but, briefly, exact matching of assembled genes in metagenomes is difficult as short read assembly results usually in a chimeric consensus of the gene variants in the sample. We have addressed the issue by using unassembled metagenomic reads and compared the nucleotide fingerprints of the samples to each other, which enables comparing the whole pools of ARGs and MGEs in the samples in a nucleotide signature rather than annotation-based manner. We have added DNA signature profile -based analyses to the manuscript and clarified when we mean gene types or DNA level analyses.

“Our study provides insight into the assembly of the infant resistome and MGEs during the first six months of life. Infants shared their resistomes and MGEs with their own mothers mediated via vertical transfer from mother’s gut and breast milk microbiota.”

Comment: I don’t think vertical transmission of MGE has been shown.

We have removed the phrase vertical transfer from the text.

“Based on our study and previous work^{8,11} it seems that infants inherit the legacy of past antibiotic use as they carry high loads of antibiotic resistant bacteria acquired from their mothers and the environment even prior to being exposed to antibiotics themselves.”

Comment: I would guess IAP would matter.

Based on our results IAP doesn't have a significant effect on how ARGs and MGEs are shared between related mothers and infants. The result has been added to the main text and also to the Supplementary Figure S6. However, we cannot exclude that with a larger set of samples, a significant stratification could be seen, but likely the effect is not very large as we could not observe it in our cohort.

"However, actions that shape the overall microbiota composition of the infant gut, such as breastfeeding or antibiotics, likely have a strong impact on the resistance gene load, as phylogeny was the major determinant of the resistome and most ARGs seem to be carried by a limited number of taxa in the infant gut."

Comment: How would IAP affect the vaginal resistome and ARG content, and what could the potential role of the vaginal microbiota be for the mother to child transmission of ARG?

We also find this question very interesting and would have liked to study this, but we were unfortunately limited by resources. We have added discussion on the effect of the vaginal microbiome to the conclusions -section.

Reviewer #2 (Remarks to the Author):

I prefer an anonymized report. Call me old school, but it is my preference.

This is a very exciting and interesting study regarding the source origin of both ARGs and MGEs in the infant gut microbiome and the breast milk microbiome.

While there are multiple exciting elements to this study, there are also some fundamental key limitations which restrict both interpretation and findings.

Exciting aspects

1. This study is inherently novel in its aims and goal of looking for the source of the neonatal/infant resistome. Namely, they sought to characterize both ARGs and MGEs in the maternal gut and breast milk microbiomes, then correlate these with the neonate/infant stool patterns. They study 16 mother-infant pairs over a span of up to 8 months, with neonates sampled at 1 month and infants sampled at 6 months. Mothers were sampled at 32 weeks gestation and one month post partum. Thus, there are unpaired maternal and infant samples at 32 weeks (moms only collected, fetal collection clearly not possible) and again at 6 months (infant only, no maternal breast milk nor stool).

Thus the sample collection is both exciting and an inherent strength but also a significant limitation. This is further discussed below. As such limitations are inherent and evident, the abstract and last paragraph of introduction must be rewritten as they overstate the design and are not reflective of what was actually collected and done.

We thank Reviewer 2 for their comment. We have modified the last part of the introduction by replacing the phrase "key factors influencing the infant gut resistome and MGEs" with "factors influencing the infant gut resistome and MGEs" and added clarification of how many of the

samples overlap by collection time to avoid overstating the design. The sampling was designed that it would allow us to determine not only the transmission of ARGs and MGEs from maternal gut on breast milk to infants, but also the effects of IAP and breast feeding on the infant gut resistome and MGEs. To be able to fulfill all of these purposes we have decided to sample at the times selected for this study. The sample collected during pregnancy was needed to be able to determine if there were significant effects on the ARG levels after the IAP treatment in the maternal gut or if the maternal baseline levels of ARGs differed.

The comment related to the changes in the abstract is unclear as we have not described samples in the abstract and at the moment the abstract does not state that we claim breast milk and maternal feces are the only sources of ARGs and MGEs, since we only state that the similarity between those of the infant's own mother and the infant gut is higher than between unrelated mothers and that it seems that infants inherit ARGs and MGEs from their mothers, which was supported by our results. However, we have modified the abstract and hope that it has improved with the changes. We have added discussion on the other possible sources of ARGs and MGEs in the results section "Infant ARG and MGE compositions resemble those of their mothers suggesting maternal effects via gene transfer" and in the discussion.

2. The methodology (if one ignores the study design limitations and concerns) is very sound and I have no concerns about the metagenomics analysis and pipelines used per se. They are very standard and appropriate. As a minor point, measuring significance in their beta diversity by PERMANOVA or Adonis would be good--it is alluded to in the text but not placed on the legend nor in Figure 2.

We have added reference to the Supplementary table 2 where the significances have been reported in the legend of Figure 2.

The exciting part is the deep sequencing of the microbiome in the human milk and the tools they used to do so. This is a relatively novel application of existing tools, and is good to see done.

3. The degree to which the milk and infant gut harbor ARGs and MGEs is a key contribution to the literature. Although not novel (see cited Gibson paper) this is well done and from a cohort of term, vaginally delivered infants.

That said, the lack of other maternal and infant body sites, including placenta and oral and skin, is a real missed opportunity and limits all source and sink discussions. This is further confounded by the fact that the maternal gut microbiome is known to change in pregnancy, but when it resumed pre-pregnant community structure and function is not yet known. Assuming a maternal composition at 32 weeks and 1 month will be representative of 6 months is not appropriate.

We have addressed these concerns below in Limitations 1.

Limitations

1. Sample collection is actually unpaired and quite limited. Trying to trace the source of

neonatal/infant ARGs and MGEs with nothing but 1 week and 1 month maternal breast milk and 32 week and 1 month stool is inherently problematic and the fundamental limitation to the study. Since the infant stool is collected at one and six months, there is actually only a single temporal coincident time point (1 month) and while valuable is still a significant limitation.

Beyond the temporal limitations, there is also concern for tracing source only from breastmilk and stool. The investigators ignore the data, notably regarding E. coli and strep and klebsiella, from multiple labs demonstrating a placental microbiome. They similarly neglect studies from multiple labs looking at the early onset diversity of multiple infant body sites, their differences, and sources from placenta and maternal stool, breastmilk, vagina, and oral communities. In a nutshell, making huge source assumptions looking only at unpaired maternal milk and stool is a tremendous leap. At present, the conclusions are not supported by the presented data.

Our aim was to investigate if mothers transmit ARGs and MGEs to their infants, not necessarily to study which parts of the mother's microbiome are most important. We decided to sample breast milk and gut, since breast milk is likely a main source of digested microbes for the infant due to it being the main nutrition for breastfed infants and since the gut microbiota of mothers is more similar in its composition to the infant gut than the composition of microbiota of any other body sites. Thus, our assumption was that by sampling those two sites, that we suspected would be predominant sources of microbes to the infant gut in terms of numbers of bacteria, we would be able to see whether maternal ARGs and MGEs are shared with infants.

We agree that our study cannot state that the maternal gut and breast milk are the sole, or even the most important, sources since the sampling was not designed to test this hypothesis. We agree that sampling other body sites besides gut and breast milk would add valuable information in which of the maternal sources are most influential in the development of the infant gut resistome. Conducting such a study would be an intriguing possibility now that we have seen evidence of shared ARGs and MGEs between infants and mothers. We have added discussion regarding the transfer of genes from other body sites to the result section describing the results of the Venn diagram analysis and to the discussion.

In our view, treating sample pairs from the same family statistically as unpaired would not be justified, as they indeed are pairs, although we of course agree that the mother-infant samples are not always sampled at the same time or represent the same type of sample. Based on the concerns raised by Reviewer 2, we did new analyses to reveal whether there are differences in the similarity of mothers to their own infants' samples collected at the same (at the one-month timepoint) or different times (antepartum maternal sample compared to six-month-old infants or one-month-old infants and the one-month postpartum maternal sample compared to six-month-old infants). The new analysis has been added to the new Figure S5 and the main text. We discovered that there were no significant differences in how similar the mothers are to their infants dependent on if the samples were collected at the overlapping time point of one month or at different time points. Thus, our conclusion is that the microbiome, resistome and MGEs of mothers are stable enough to detect the signature of transmitted ARGs and MGEs in the infant even when the sampling has occurred at different times. However, it is likely that time of sampling plays a role, but that the changes are in the species, resistome and MGE composition are not so

dramatic that the signature of the individual would be lost completely. Moreover, we did not observe any significant differences in the clustering of microbiomes, resistomes and MGEs of mothers' ante- and postpartum samples or in one and six-month-old infants (Supplementary table 2, ADONIS) further supporting that the changes occurring between these samples are minor and do not cause conflict in analyzing transmission between mothers and infants.

Based on the new analyses and our previous results, we would like to disagree with Reviewer 2 and argue that our results are supported by our analyses and discussion related to the transmission of genes between mothers and infants is also supported by results and literature. Based on the comments we have added discussion related to other maternal sources of ARGs and MGEs to the to the section describing the result of the Venn diagram analysis and to the discussion.

3. The figures would benefit from some editing. The axis labels are not particularly helpful and presenting box and whiskers of Shannon diversity without other alpha measures (Chao 1, bet deiversiy) is a key limitation. The data in Figure 1 is really not novel nor well done, and do not heighten enthusiasm for the study and its findings.

Figure 1 has been edited to only include the main findings and Chao1 estimates have been added to the supplementary figure Figure S1. We have improved the quality of figures.

4. Figure 4 is inherently flawed for all the reasons discussed above. Running BC dissimilarity on actually non-paired samples and without other body sites is disingenuous and the conclusions cannot be considered valid as they were never actually tested. The same applies to Figure 5 and Figure 6.

We have addressed the points raised in this comment previously in the answer to Limitations 1. Analyses in Figure 4 and 5 are based on similarity matrixes, which contain similarity values computed for each sample to sample comparison. This type of analysis does not require the samples to be paired, since the distances are computed independently of assumptions of paired samples. We assume that Figure 6 is not problematic in this sense as it is not related to transmission between mothers and infants but to the effects of IAP and breastfeeding.

5. A final real missed opportunity and neglected key point is the number of subjects on probiotic supplements. This is crucially important, since this same team of investigators has previously published data suggesting maternal to infant transfer of probiotics occurs in utero and during lactation. However, there is literally no demonstration of this spp and strains in their metagenomic analysis in this paper. This renders the formal possibility that their unaccounted for ARGs and MGEs could actually be arising from these probiotics. This must be tested, since 10/16 mothers were on them.

We thank Reviewer 3 for their comment. We have purposely refrained from analyzing the transmission of probiotic strains in this study, as it has been studied previously and it is not in the scope of the paper and would require other methods besides metagenomics. Regarding the contribution of probiotics on the resistome and MGE transmission, based on our results described in the manuscript, the probiotic species correlated negatively with the ARGs and MGEs or did not

correlate significantly with the genes (Supplementary table 3), and thus likely do not significantly contribute to the transmission of resistance genes and MGEs from mother to infants. The probiotic species were not detected in breast milk, most possibly due to the low sequencing depth in milk. We also tested if there are differences in the abundance or composition of microbiomes, MGEs and ARGs due to probiotics and observed no significant differences further supporting the assumption that the probiotic strains did not contribute significantly to the resistome. We have added this analysis in the manuscript main text.

Reviewer #3 (Remarks to the Author):

The major findings of this paper relate to how the microbiomes, resistomes, and mobilomes of infants are shaped by their mother and by breast milk. There are several novel findings in the paper and this will be of interest to others in the field. The statistics appear to be well chosen and applied, convincing the reader that the claims are valid.

However, the manuscript is very wordy and can be shortened. Many comments in the text seem to have no greater purpose or implication. For example:

"Transposon Tn916 associated genes were found in all sample types but were especially common in breast milk and in one-month-old infants."

Why was that specific MGE chosen for comment? There are other MGE that are also found in all sample types but more common in some than others (as seen in the Figure).

We have removed the text regarding Tn916 and have shortened the text where we found that we could do so without losing valuable content.

Some other comments:

Breast milk, as the primary source of nutrition during the first months, shapes the infant gut microbiota^{14,15}, but its role in contributing to the resistome is virtually unexplored.

There have been studies, (e.g. <http://msystems.asm.org/content/3/1/e00123-17>) that have explored how breast milk vs formula feeding affects the resistome.

We thank Reviewer 3 for the comment. The citation has been added to the introduction.

Figure 1- please make the letters referring to each part of the figure panel larger. Also, to avoid confusion, when referring to parts of the panel in the text, include the letter -- i.e. use Figure 1E rather than Figure 1 in

"The relative abundances of ARGs were similar in breast milk and fecal samples from six-month-old infants and mothers ($p > 0.05$, Figure 1, Supplementary Table 2)."

We thank Reviewer 3 for their comment. We have made the letters in Figure 1 panels larger and have modified the text to include reference to the panels.

"The results suggest that early termination of breastfeeding might have negative health effects for infants due to an increased resistance potential of the gut microbiota against certain antibiotics."

Authors may want to comment on why this is a negative effect. In the introduction, the authors discuss the negative aspects of infectious pathogens carrying ARGs, but has it ever been shown that higher ARG in the microbiota leads to negative health effects?

We thank Reviewer 3 for the comment. We have modified the text to be more precise regarding the negative health effects caused by increased resistance potential. “The results suggest that early termination of breastfeeding might have negative health effects for infants due to an increased resistance potential of the gut microbiota against certain antibiotics, and, thus, likely also selecting for antibiotic resistant opportunistic pathogens able to cause infections under the right circumstances.”

REVIEWERS' COMMENTS:

Reviewer #1 (Remarks to the Author):

I think you adequately addressed my concerns related to the first version of the ms, and I agree with the interpretations and conclusions in the current version.

Reviewer #2 (Remarks to the Author):

Thank you for the opportunity to rereview this manuscript. I am very appreciative of all the work the authors undertook in order to address both mine and the other reviewers concerns.

I do wish to request they address a few very simple additional modifications:

1. IAP: I don't believe that they actually meant to state that in half of all cases they were attempting to prevent maternal-child vertical transmission of pathogens. The only three cases of such approaches clinically are antiretrovirals for prevention of vertical viral transmission, spiramycin for prevention of toxo transmission, and malarial suppressive therapy for placental malarial transmission.

In all other common instances, they are attempting to either treat maternal infection OR prevent pathobiont transmission (i.e., largely Group B strep). If it is chorioamnionitis, they are actually treating maternal infection. In the case of GBS, it is a pathobiont since it causes no disease in the mom and in 1:8000 or so cases may cause early invasive GBS disease in the first 6 days of life in the neonate.

2. In the first comment of Reviewer 2, Reviewer 2 stated "the abstract and first paragraph of the introduction must be rewritten". I believe that the authors did not note first paragraph of intro. I appreciate the changes they made in the abstract and ask some attention be similarly placed on first paragraph of intro to fully assure they have not overstated their findings.

3. The addition of Figure S5 is excellent and appreciated. It thus reinforces the point that Reviewer 2 was making--the maternal-child transmission is relatively time independent, which suggests it occurs in utero and very shortly thereafter. It is also true it appears stable, although no such community modeling nor MCM analysis was performed.

I would concur that given their new data, they do have better support for their conclusions than they did initially and are pleased to see the study strengthened.

We would like to thank the Reviewers for giving additional comments. Here are our responses to the review comments.

Reviewer #2 (Remarks to the Author):

Thank you for the opportunity to rereview this manuscript. I am very appreciative of all the work the authors undertook in order to address both mine and the other reviewers concerns.

I do wish to request they address a few very simple additional modifications:

1. IAP: I don't believe that they actually meant to state that in half of all cases they were attempting to prevent maternal-child vertical transmission of pathogens. The only three cases of such approaches clinically are antiretrovirals for prevention of vertical viral transmission, spiramycin for prevention of toxo transmission, and malarial suppressive therapy for placental malarial transmission.

We thank Reviewer 2 for their comment. We have modified the text and removed speculation relating to IAP preventing the transmission of species.

In all other common instances, they are attempting to either treat maternal infection OR prevent pathobiont transmission (i.e., largely Group B strep). If it is chorioamnionitis, they are actually treating maternal infection. In the case of GBS, it is a pathobiont since it causes no disease in the mom and in 1:8000 or so cases may cause early invasive GBS disease in the first 6 days of life in the neonate.

2. In the first comment of Reviewer 2, Reviewer 2 stated "the abstract and first paragraph of the introduction must be rewritten". I believe that the authors did not note first paragraph of intro. I appreciate the changes they made in the abstract and ask some attention be similarly placed on first paragraph of intro to fully assure they have not overstated their findings.

We thank Reviewer 2 for their comment. We have modified the last, (which is the paragraph that discusses study design, not the first paragraph) paragraph of the intro and hope it is now more descriptive of what was studied and does not overstate the design.

3. The addition of Figure S5 is excellent and appreciated. It thus reinforces the point that Reviewer 2 was making--the maternal-child transmission is relatively time independent, which suggests it occurs in utero and very shortly thereafter. It is also true it appears stable, although no such community modeling nor MCM analysis was performed.

I would concur that given their new data, they do have better support for their conclusions than they did initially and are pleased to see the study strengthened.

We thank Reviewer 2 for their feedback.